# A chromosome-level *Camptotheca acuminata* genome assembly provides insights into the evolutionary origin of camptothecin biosynthesis

Minghui Kang[1], Rao Fu [1], Pingyu Zhang[1], Shangling Lou[1], Xuchen Yang[1], Yang Chen[1], Tao Ma [1], Yang Zhang [1], Zhenxiang Xi [1] & Jianquan Liu [1✉]

Camptothecin and its derivatives are widely used for treating malignant tumors. Previous studies revealed only a limited number of candidate genes for camptothecin biosynthesis in *Camptotheca acuminata*, and it is still poorly understood how its biosynthesis of camptothecin has evolved. Here, we report a high-quality, chromosome-level *C. acuminata* genome assembly. We find that *C. acuminata* experiences an independent whole-genome duplication and numerous genes derive from it are related to camptothecin biosynthesis. Comparing with *Catharanthus roseus*, the loganic acid *O*-methyltransferase (LAMT) in *C. acuminata* fails to convert loganic acid into loganin. Instead, two secologanic acid synthases (SLASs) convert loganic acid to secologanic acid. The functional divergence of the *LAMT* gene and positive evolution of two *SLAS* genes, therefore, both contribute greatly to the camptothecin bio-synthesis in *C. acuminata*. Our results emphasize the importance of high-quality genome assembly in identifying genetic changes in the evolutionary origin of a secondary metabolite.

[1] Key Laboratory of Bio-resource and Eco-Environment of Ministry of Education, College of Life Sciences, Sichuan University, Chengdu, China. ✉email: liujq@nwipb.ac.cn

Malignant tumors pose a serious threat to human health, and chemotherapy is often given as an adjuvant treatment following surgery or radiation. In 1966, camptothecin, a monoterpene indole alkaloid (MIA), was discovered to be an anti-tumor agent[1]. It was isolated from the wood and bark of *Camptotheca acuminata* Decne., which belongs to a monotypic genus endemic to southwestern China. Camptothecin selectively binds to topoisomerase I and prevents the re-ligation of the DNA strands, and this effectively inhibits the proliferation of tumor cells[2–5]. Camptothecin (including its derivatives) is therefore widely used as a chemotherapeutic drug for treating malignant tumors[6], and has become the second most important plant-derived anti-cancer agent after taxol[7–9]. In spite of its great economic value[10], it is still poorly understood how the unique camptothecin biosynthetic pathway has evolved in *C. acuminata*.

On the basis of annotated metabolites and enzymes identified[11], the camptothecin biosynthetic pathway in *C. acuminata* is similar to that of some other MIAs, such as vinblastine and vincristine in *Catharanthus roseus* (L.) G.Don. Both pathways synthesize complex intermediate organic molecules (e.g., geraniol, 8-oxogeranial[12–14], iridodial[15], 7-deoxyloganetic acid[16], 7-deoxyloganic acid, and loganic acid[17]) using the same set of enzymes, which are presumably encoded by homologous genes. After these intermediate stages, however, the two biosynthetic pathways are quite different from each other. In *C. roseus*, loganic acid is first converted into loganin by an *S*-adenosyl-L-methionine-dependent carboxyl methyltransferase (loganic acid *O*-methyltransferase, LAMT)[18,19], and then into secologanin by secologanin synthase (SLS)[20,21]. In contrast, loganic acid is converted directly to secologanic acid and then to strictosidinic acid in *C. acuminata*[11]. Thus, it is clear that the process by which loganic acid is converted to secologanic acid has been critical in the evolution of the highly effective biosynthesis of camptothecin in *C. acuminata*.

Here, we sequence and assemble the whole genome of *C. acuminata* base on single-molecule real-time long reads from the Pacific Biosciences (PacBio) Sequel platform, and construct pseudo-chromosomes using high-throughput chromosome conformation capture (Hi-C) techniques[22,23] base on a chromosome number of $2n = 42$[24]. We identify an independent whole-genome duplication (WGD) event in this high-quality chromosomal-scale genome. Then we use this reference genome to identify genes associate with the evolutionary origin of camptothecin biogenesis by means of comprehensive homology searching, gene family analyses, co-expression analyses of RNA-seq datasets and functional verification of site mutations in key homologous genes. We aim to address how camptothecin biosynthesis has evolved by conversion of the same intermediate chemical into different subsequent products comparing with vinblastine/vincristine biosynthesis in *C. roseus*[14,25].

## Results

**Improvement of *C. acuminata* genome assembly.** The previously published *C. acuminata* genome assembly (Cac genome assembly v2.4) was created with the ALLPATHS-LG assembler using Illumina short-read technology, producing an assembly of 1,394 scaffolds (5,219 contigs) spanning 403.2 Mb, with a scaffold N50 of 1.75 Mb (contig N50 = 194.6 Kb)[26]. To improve this short-reads-based assembly, we re-sequenced and assembled the genome of *C. acuminata* using single-molecule real-time (SMRT) sequencing technology from Pacific Biosciences (PacBio), thus improving the accuracy of the assembly obtained from the Illumina platform. We then connected the contigs into pseudo-chromosomes using Hi-C techniques (Supplementary Table 1). The final 414.95 Mb genome assembly (*C. acuminata* assembly V3.0) contained 1,130 contigs (contig N50 = 1.47 Mb), with a maximum length of 4.71 Mb; it is close to the estimated genome size of 404.95 Mb based on K-mer analysis (Supplementary Table 4). Using Hi-C data, 393.99 Mb (96.03%) were anchored onto 21 pseudo-chromosomes, which were ordered and oriented; the scaffold N50 was 18.28 Mb and the maximum chromosome length was 39.28 Mb (Table 1 and Supplementary Fig. 3). The mapping rate and coverage of the assembled *C. acuminata* sequences were estimated to be, respectively, 94.84% and 95.71% using Illumina short reads (Supplementary Table 2). These results indicated that the assembled sequences had high base accuracy, high continuity, and a high degree of genome coverage.

The completeness of the genome assembly was quantified using a large core set of highly conserved plant-specific single-copy orthologs and the number of intact long-terminal repeat retrotransposons (LTR-RTs). Of the 1,440 plant-specific orthologs, 1,388 (96.4%) were identified in the assembly, and 1,367 (94.9%) of them were considered to be complete; the figure for the previously published genome assembly was 93.6% (Supplementary Table 5). Our genome assembly was found to have more intact LTR-RTs (465) than the previous assembly (340) (Supplementary Fig. 4 and Supplementary Table 6). We also found, through comparison using LAST, that the PacBio long-read assembly corrected assembly errors and improved the continuity of the Illumina short-read assembly, by filling gaps and correcting low-quality sequences surrounding gaps and base call errors (Supplementary Fig. 5a). All the above results suggested that ours is a higher quality genome assembly than that previously published.

**Table 1 Statistics for Cac genome assembly v2.4 and *C. acuminata* assembly V3.0.**

| | Cac genome assembly v2.4 (ALLPATHS-LG + SOAP GapCloser) | *C. acuminata* V3.0 (FALCON + Quiver + Pilon + Hi-C) |
|---|---|---|
| Sequencing platform | Illumina HiSeq 2000 | PacBio Sequel |
| Assembly size (bp) | 403,174,860 | 414,951,143 |
| GC % | 32.83 | 32.87 |
| Number of scaffolds | 1394 | 775 |
| Scaffold N50 size (bp) | 1,751,747 | 18,276,129 |
| Scaffold N90 size (bp) | 431,274 | 12,137,589 |
| Number of contigs | 5219 | 1130 |
| Contig N50 size (bp) | 194,584 | 1,473,707 |
| Contig N90 size (bp) | 50,104 | 351,195 |
| Gap % | 0.94 | 0.01 |
| Longest sequence length (bp) | 8,423,530 | 39,282,906 |

**Repeat and gene annotations**. Repetitive sequences were identified using a combination of ab initio and homology-based approaches. In total, 37.55% of the assembled sequences were annotated as repetitive, including 18.41% of retrotransposons and 6.86% of DNA transposons. Long-terminal repeat (LTR) retrotransposons were found to account for 13.50% of the genome (Supplementary Table 7).

We annotated protein-coding genes using a pipeline combining RNA-seq data-based and ab initio-based evidence. We predicted a total of 27,940 genes, of which 9,045 had alternatively spliced transcripts, with an average transcript length of 8,394 bp, a coding sequence size of 1,657 bp, and a mean of 6.14 exons and 1.85 transcript per gene (Supplementary Table 8). Overall, functions were assigned to 27,122 genes (97.07%), based on their homologies to annotated proteins in SwissProt and TrEMBL database. Further functional annotation using InterProScan estimated that 95.93% of the genes contained conserved protein domains. 88.38% of the genes could be classified with Gene Ontology (GO) terms, using the combined results from Blast2GO and InterProScan, and 33.36% mapped to known plant biological pathways based on the KEGG Pathway database (Supplementary Table 9).

By comparing the annotation of gene structure between our *C. acuminata* V3.0 assembly and Cac genome assembly v2.4, we found that we had annotated fewer, but on average longer, genes (27,940) than in the previous assembly (31,825), with on average more exons per gene (6.14 compared to 5.07) (Supplementary Table 8). We found that in the Illumina-based genome assembly, some genes were partially annotated due to gaps and low-quality sequences surrounding gaps and some were falsely annotated due to misplaced assembly (Supplementary Figs. 5b, c and 6). To explain the improvement in gene structure annotation, we did an all-to-all blast between our genes and those previously published, and found that 2,446 genes in the previous assembly could not be found in the all-to-all blast results and 2,237 of them did not have any functional annotation (Supplementary Data 1). A box-plot shows that these genes are significantly shorter than the other genes in the previous assembly (significance tested by Wilcoxon method with $p$-value $< 2.2e^{-16}$) (Supplementary Fig. 7a). This suggests that these genes may have been falsely annotated. Based on the result of MCScanX analysis, we found 18,116 1:1 orthologous gene between these two assemblies, and 12,534 of them were significantly longer in our assembly than in the previous assembly (significance tested by the Wilcoxon method with $p$-value $< 2.2e^{-16}$) (Supplementary Fig. 7b and Supplementary Data 1). This indicates that these genes may be partially annotated. Furthermore, there are 355 genes in *C. acuminata* V3.0, which have two or even three matches in Cac genome assembly v2.4 according to MCScanX, but combining these results with those of LAST assignment, there is only one correct match for each of these genes (Supplementary Data 1). All the above findings suggest that our PacBio-based genome assembly has allowed for more accurate gene structure annotation.

**Phylogenetic and whole-genome duplication analyses**. We clustered the annotated genes into gene families for *C. acuminata* and seven other plant species. A total of 23,047 *C. acuminata* genes (82.49%) clustered into 13,188 gene families, which included 7,434 (56.37%) gene families shared by all 8 species and 413 (3.1%) *C. acuminata*-specific families (Supplementary Table 10 and Supplementary Fig. 8). Our gene ontology (GO) term enrichment analysis ($p < 0.05$, FDR $< 0.05$) revealed that those genes unique to *C. acuminata* were involved in the negative regulation of the seed dormancy process, release of seeds from dormancy, positive regulation of the gibberellic acid mediated signaling pathway, positive regulation of seed germination and so on, suggesting that they may play important roles in seed dormancy and germination (Supplementary Table 11). This may be related to the fact that the seed of *C. acuminata* has a 4-month dormancy period[27].

We selected 2,025 single-copy gene families among the 8 species to construct a phylogenetic tree, which showed that *C. acuminata* and *R. delavayi* lay on a branch outside the euasterids and at the basal position of the asterids. We estimated that *C. acuminata* and *R. delavayi* diverged from other euasterids around 118 million years ago (Mya) (107–124 Mya), and that *C. acuminata* and *R. delavayi* diverged around 107 Mya (92–115 Mya) (Fig. 1a). These data indicate that these two species have a closer genetic relationship with each other than with the other species included in the tree, consistent with their previously assigned phylogenetic placement[28].

It is now widely recognized that WGD events have a major impact in shaping plant genome evolution and speciation. We used the distribution of synonymous substitution rates per gene (Ks) between collinear paralogous genes to identify WGD events, based on the assumption that the number of silent substitutions per site between two homologous sequences increases in a relativity linear manner with time. A total of 3,213 syntenic blocks, containing 33,904 pairs of collinear genes, were identified in the *C. acuminata* genome. The total length of these syntenic blocks was 315.1 Mb (84.60% of the assembly), suggesting that the majority of the *C. acuminata* genome was duplicated during its evolution. The Ks values for the collinear gene pairs peaked at 0.40–0.44, corresponding to an ancient WGD event that occurred around 67.11–73.83 Mya in the *C. acuminata* lineage (and possibly throughout the Cornales) after its initial split from the asterids (Fig. 1b). Synteny analyses comparing the genomes of *C. acuminata* and *V. vinifera* also showed clear evidence of a single WGD event in the *C. acuminata* lineage. For each genomic region in *V. vinifera*, we typically found two matching regions in *C. acuminata* with a similar level of divergence (Fig. 1c and Supplementary Fig. 9). The genome of *V. vinifera* has not undergone any recent WGDs after the hexaploidization event shared by core eudicots[29]. The overall 2:1 syntenic relationship between *C. acuminata* and *V. vinifera* suggested that *C. acuminata* experienced another WGD event after its divergence from *V. vinifera*. This WGD occurred independently of all previously reported asterid-specific WGD events.

The expansion and contraction of gene families play critical roles in phenotypic diversification in plants[30]. Plants with duplicated genes are predicted to have more adaptive power than those with single copies, due to the likely gain of novel functions and pathways. We conducted expansion and contraction analysis of 13,346 shared gene families based on the phylogenetic tree we constructed, and discovered 2,951 expanded and 1,733 contracted families in *C. acuminata* relative to *R. delavayi* (Fig. 1a). Based on the MCScanX result, 9,249 genes in 2,951 expanded families were further divided into five classes based on their origins and locations: singleton (0.34%, 32), dispersed (8.76%, 810), proximal (2.74%, 253), tandem (7.51%, 695) and WGD/segmental (80.64%, 7,458) (Supplementary Table 12). This may suggest that WGD and tandem duplication were the main processes contributing to the expansion of gene families in *C. acuminata*. GO enrichment analysis of the expanded gene families indicated that these genes were enriched for flower and pollen development, ion transport, and some metabolic and biological processes including the indole biosynthetic process, while the tandemly repeated genes were enriched for the indole biosynthetic process and glucosyltransferase function, which are important in the camptothecin biosynthesis pathway (Supplementary Tables 13, 14).

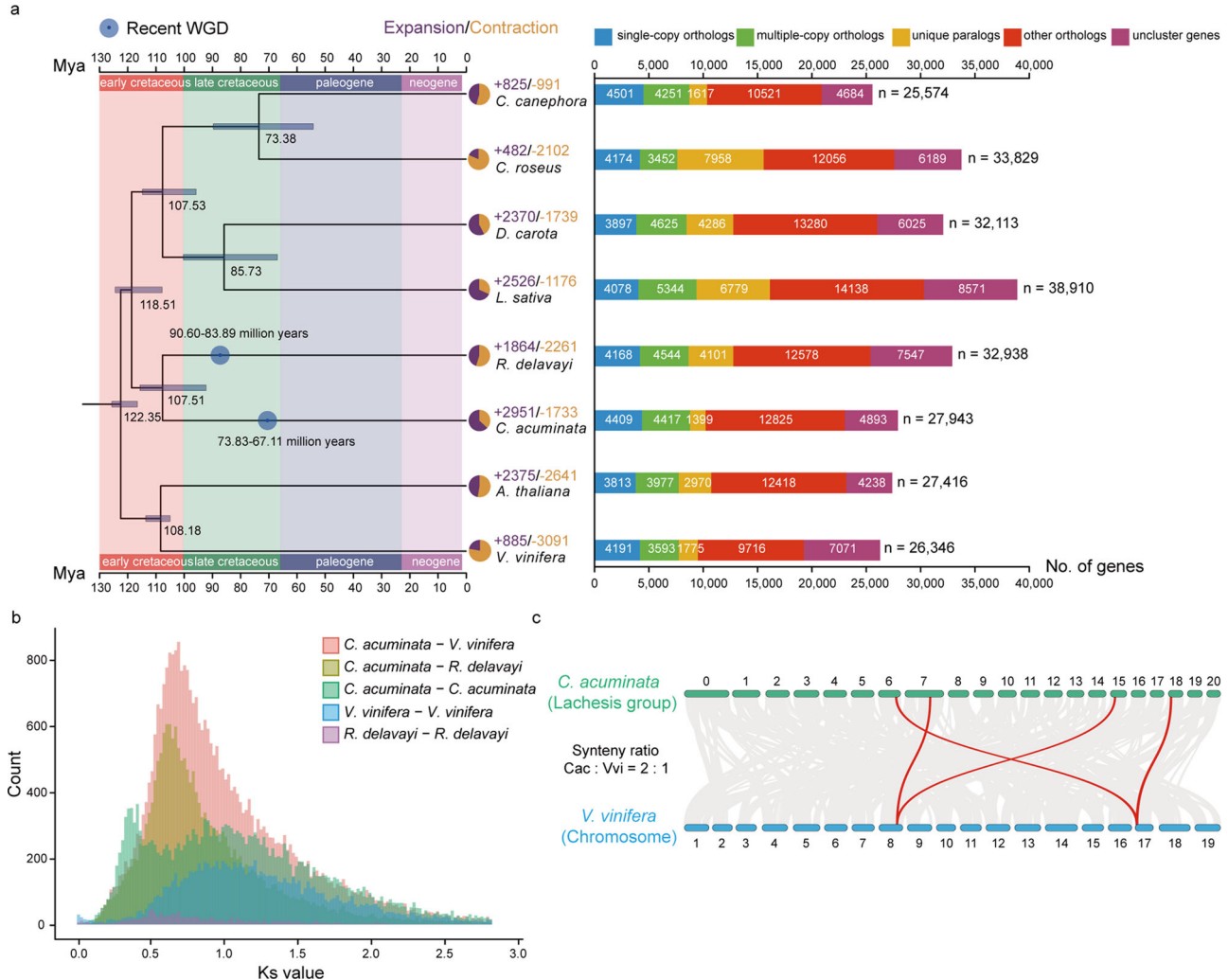

**Fig. 1 Phylogenetic analysis of the *C. acuminata* genome. a** Phylogenetic tree for *C. acuminata* and seven other plants. Genes of *C. acuminata* and other sequenced genomes are classified into five classes and the direct numbers (*n*) are shown on the right with bar charts. All branch bootstrap values are 100. Gene family expansions are indicated in purple, and gene family contractions in light-brown; the corresponding proportions among the total changes are shown using the same colors in the pie charts. The estimated divergence time (million years ago, MYA) is indicated at each node; bars are 95% confidence intervals (CI) (each center is defined as mean value). Circles in blue represent recent whole-genome duplication (WGD) events. **b** Ks values revealed a recent WGD event during the evolution of *C. acuminata* and a WGD event shared by *C. acuminata* and *V. vinifera*. **c** Collinear relationship between *C. acuminata* and *V. vinifera* chromosomes. The collinearity pattern shows that typically an ancestral region in the *V. vinifera* genome can be traced to two regions in *C. acuminata*. Gray bands in the background indicate syntenic blocks between the genomes spanning more than 15 genes; some of the 1:2 blocks are highlighted in red. Source data underlying Fig. 1a, b are provided as a Source Data file.

**Identification of key genes involved in camptothecin biosynthesis.** The biogenesis of camptothecin in *C. acuminata* is of particular interest because it is considered to be a major source of this anti-tumor alkaloid. Analysis of genes from available *C. acuminata* transcriptome data has already enabled successful identification of some genes for camptothecin biosynthesis[11,31]. However, most previous analyses utilized de novo transcriptome assemblies, which do not provide a full dataset of the *C. acuminata* genes (Complete BUSCOs: 1,240/1,440 (86.1%), caa_assembly_v_10072011 downloaded from Medicinal Plant Genomics Resource (MPGR) (http://medicinalplantgenomics.msu.edu/)) (Supplementary Table 15). Our high-quality genome sequence and annotation dataset for *C. acuminata* adds more candidate genes and gene families potentially involved in camptothecin biosynthesis. Based on the results of homology searching and functional annotation, we identified candidate genes that may encode 6 enzymes for tryptamine synthesis and 18 enzymes for the indole alkaloid biosynthesis pathway and further determined other members of their gene

families (Supplementary Tables 16 and 17). We outlined the putative camptothecin biosynthetic pathway based on the KEGG database, previously published results[11] and the expression profiles of each candidate gene in 15 tissues (Fig. 2a). Previous studies on *C. roseus* showed that two members of the CYP450 subfamily CYP72A, secologanin synthase (SLS/CYP72A219) and 7-deoxyloganic acid 7-hydroxylase (7-DLH/CYP72A224), are critical in indole alkaloid biosynthesis[17,19,21]. In order to further identify potential genes encoding 7-DLH and SLS-like enzymes (also called secologanic acid synthase (SLAS) in a previous study[11]) in *C. acuminata*, we constructed phylogenetic trees of all candidate genes identified here that may encode these enzymes in the subfamily CYP72A together with two previously published SLS and 7-DLH sequences in *C. roseus*. We found that two genes identified here, *CacGene13171* and *CacGene10832*, clustered with the previously reported *7-DLH* gene while the other two, *CacGene13172* and *CacGene10833*, clustered with the previously reported *SLS* gene with high statistical support values (Fig. 2c).

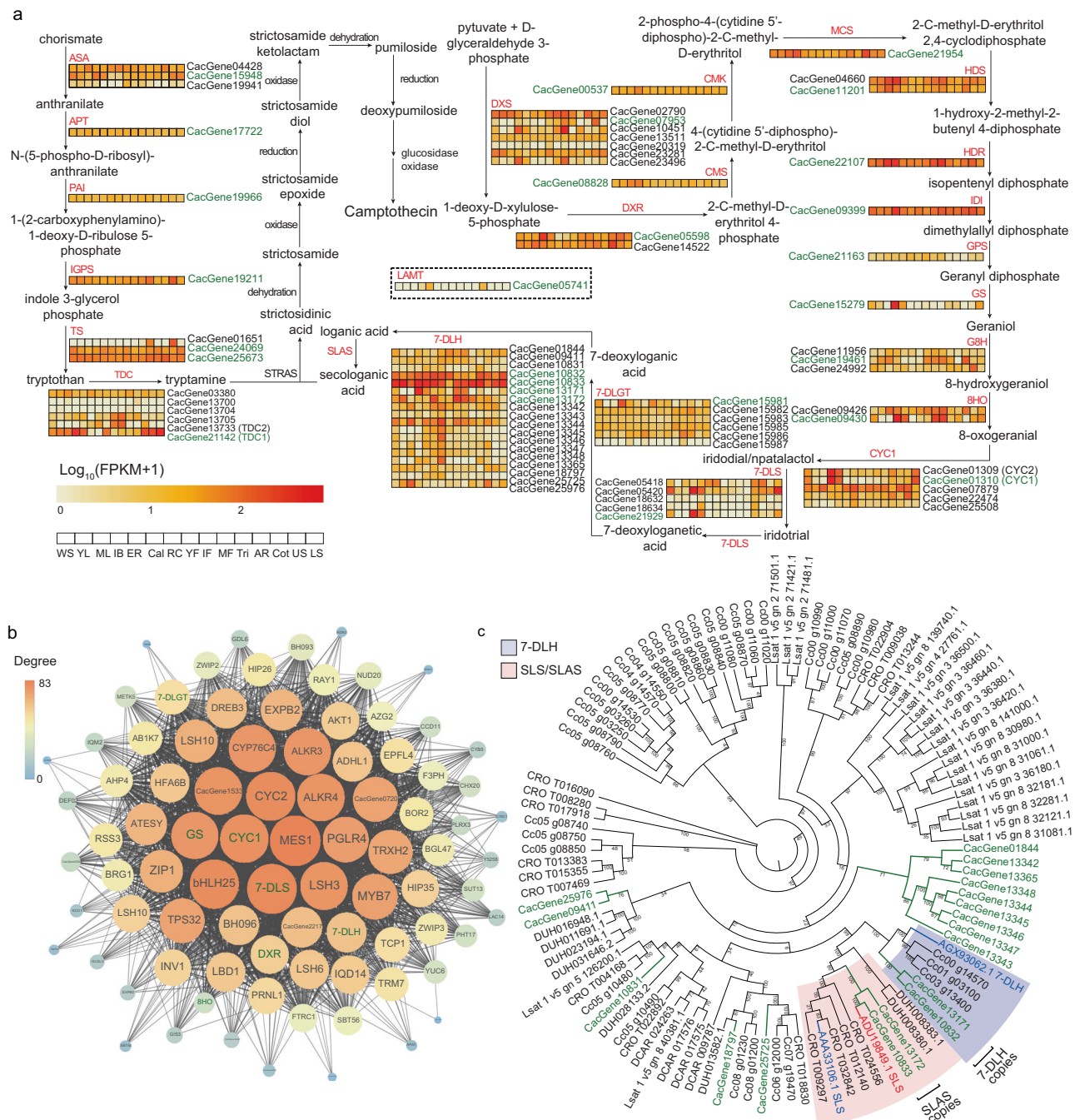

**Fig. 2 Genes involved in camptothecin biosynthesis. a** A simplified representation of the camptothecin biosynthetic pathway. Top hits for pathway genes identified by blast and pathway genes in the co-expression network are highlighted in green. The expression value for each gene is indicated in color on a log$_{10}$(FPKM + 1) scale for fifteen tissues: whole seedlings (WS), young leaf (YL), mature leaf (ML), immature bark (IB), entire root (ER), callus (Cal), root culture (RC), young flower (YF), immature fruit (IF), mature fruit (MF), trichomes (Tri), advance roots (AR), cotyledons (Col), upper stem (US), lower stem (LS). **b** The WGCNA "grey60" module related to camptothecin biosynthesis as represented by a node and edge graph. Connection strength is represented by edge width (edge weights < 0.25 are omitted). **c** A phylogenetic tree of all candidate genes in the *CYP72A* subfamily. The sequences shown in blue and red indicate previously published *7-DLH* and *SLS* genes in *C. roseus* while those in green show candidate genes identified in the *C. acuminata* V3.0 genome.

Base on re-analysis of previously reported transcriptomic data[31] using the newly sequenced genome as reference, we calculated levels of expression for all candidate genes in 15 tissues and found that almost all pathway-related genes were expressed, and most expanded families had more than one highly expressed member (Fig. 2a, Supplementary Data 2 and Supplementary Fig. 10). Expansion and expression of these gene families may, therefore, have contributed to the evolution of camptothecin

biosynthesis in *C. acuminata*. To better examine co-expression of the pathway-related genes, we did differential gene expression analysis and constructed weighted gene co-expression networks with WGCNA using the differentially expressed genes and obtained 30 clusters (Supplementary Figs. 11–14 and Supplementary Data 3–4). Interestingly, some important gene copies from the seco-iridoid pathway were grouped into the "grey60" module output by WGCNA; they included *CacGene15279* (for

geraniol synthase/GS), *CacGene01310* (for iridoid synthase/IS), *CacGene21929* (for 7-deoxyloganetic acid synthase/7-DLS), *CacGene15981* (for 7-deoxyloganetic acid *O*-glucosyltransferase/7-DLGT), *CacGene13171* (for 7-deoxyloganic acid 7-hydroxylase/7-DLH) and other related genes (Fig. 2b and Supplementary Table 18). In addition, we found that most of these genes involved in the seco-iridoid pathway encode cytochrome P450 (CYP450) oxygenases, and the downstream steps after strictosamide also involve a set of enzymes responsible for reduction and oxidation[11] (Fig. 2a), which may indicate that *CYP450* genes play an important role in the biosynthesis of camptothecin, as they do in other plant terpene pathways[32,33]. In order to identify candidates for the later steps in camptothecin biosynthesis, we therefore screened candidate *CYP450* genes in *C. acuminata* by homology searching and structural domain alignment, and constructed their phylogenetic relationships. Then we clustered them into families and subfamilies based on domain annotation, functional descriptions of the homologs in *A. thaliana* and the SwissProt database (Supplementary Fig. 15). These genes may be involved in camptothecin biosynthesis, although verification of the functions of the corresponding proteins are needed to determine which steps they are involved in. We have therefore extended the set of candidate genes for camptothecin biosynthesis based on our chromosome-scale genome assembly and gene annotation for *C. acuminata*.

**Branching of the camptothecin biosynthesis pathway from that of vinblastine/vincristine**. Our homology searches and previous studies[11,18,19] suggested that *C. acuminata* used genes in camptothecin biosynthesis similar to those participating in the production of vinblastine/vincristine in *C. roseus* up to the point at which loganic acid is produced (Fig. 3a). However, in *C. roseus* loganic acid is transformed to loganin by loganic acid *O*-methyltransferase (LAMT) and further to secologanin by secologanin synthase (SLS), whereas in *C. acuminata* it is converted directly to secologanic acid by an SLS-like enzyme, secologanic acid synthase (SLAS)[11,25]. We therefore explored the genetic changes underlying this biosynthetic differentiation. We firstly examined the chemical structures of the intermediate products in the camptothecin and vinblastine/vincristine biosynthetic pathways. LAMT adds a methyl group to loganic acid to produce loganin while SLS and SLAS implement similar ring opening reactions in loganin or loganic acid to produce secologanin or secologanic acid respectively. The major difference between secologanin and secologanic acid is the presence or absence of a methyl group, which is added by LAMT in *C. roseus* (Fig. 3a). It is therefore likely that CaLAMT fails to add this methyl, while SLAS directly converts loganic acid to secologanic acid, which has led to the differentiation of the two contrasting biosynthesis pathways.

To test this hypothesis, we firstly identified candidate *LAMT* genes in the *C. acuminata* genome. Based on homology searching, functional annotation and gene family construction using our high-quality protein-coding gene set, we identified only one potential *LAMT* gene (*CacGene05471*) in *C. acuminata* (Fig. 2a and Supplementary Tables 16 and 17). We cloned this gene and the previously published *LAMT* gene (*KF415116*) from *C. roseus* and assayed the enzyme activities of the proteins they encoded[17]. We found that CaLAMT failed to produce loganin by adding a methyl group to loganic acid whereas CrLAMT did so (Figs. 3b, 4d, Supplementary Figs. 16–18, and Supplementary Data 7). In order to find out the key genetic changes underlying this functional shift, we constructed the protein structure of CaLAMT based on the previous published template[34] (PDB ID: 6C8R) and calculated binding energies. Five amino acid mutations between CaLAMT and CrLAMT (G240, A241, H245, Q273, and Q316)

were found to be likely accounted for the functional difference because the ligand binding related hydrogen bonds are reduced in the former, therefore, it may fail to bind loganic acid stably (binding energy greater than 0) (Fig. 4a, b). Further protein sequence alignment and phylogenetic analyses suggested that these five mutations appeared only in *C. acuminata* while two other mutations (G242 and L243) were also found in the central binding region (240-245) of CrLAMT[34] (Fig. 4c and Supplementary Fig. 19). Selection analyses by branch-site model (BSM) with *C. acuminata* set as a foreground branch showed no site under significant positive selection (LRT *p*-value < 0.05, posterior probability > 0.95), and two-ratio branch model (BM) showed that the strength of natural selection might have been relaxed in the foreground branches (background branches *ω* < foreground branches *ω* < 1, LRT *p*-value < 0.05) (Supplementary Fig. 20 and Supplementary Data 5 and 6). In addition, mutations in two amino acids (Q273 and Q316) in *C. roseus* were previously shown to reduce CrLAMT activity[34], and here four of the other five mutations in *C. acuminata* (G240, A241, G242, and H245) were also found to greatly decrease or abolish LAMT activities based on our enzyme activity experiments (Fig. 4d, Supplementary Fig. 18, and Supplementary Data 7). Overall, six mutations that affect enzyme activity are specific to *C. acuminata*, and the phylogenetic relationships in the protein-based sequence tree were consistent with the previously published species tree[35]. The failure of CaLAMT to transform loganic acid to loganin in *C. acuminata* may therefore have resulted from a degenerating pseudo-gene or evolved as direct functional divergence between two lineages.

Meanwhile, two previously reported SLASs (CYP72A565 and CYP72A610) were found to convert loganic acid directly into secologanic acid in *C. acuminata* and they could catalyze the conversion of loganin produced by CrLAMT into secologanin, as does CrSLS in *C. roseus*[36]. These two *SLAS* genes were identified as *CacGene10833* and *CacGene13172* in the present genome by sequence alignment with high identity values and they formed a phylogenetic cluster with *CrSLS* genes (Fig. 2c and Supplementary Fig. 21). Both genes are highly expressed in all tissues although *CacGene10833* (encoding CYP72A565) is expressed more highly than *CacGene13172* (encoding CYP72A610) (Fig. 2a and Supplementary Data 2). CYP72A565 was also found to have higher enzyme activity than CYP72A610 in a previous study[36]. The MCScanX results showed that these two *SLAS* genes were in two syntenic blocks and the Ks value of the two blocks was estimated to be 0.46, close to the recent WGD peak of 0.40–0.44, suggesting that they were derived from the WGD that we detected (Supplementary Fig. 22). Differences in levels of expression of these two gene copies and different enzyme activities of the two proteins may suggest sub-functional divergence between them after WGD and that they complement one another to achieve a high level of conversion efficiency. Selection analyses with homologs of other species, setting *CacGene10833* and *CacGene13172* as the foreground branch, showed that 23 different sites in these two *SLAS* genes had significant positive selection signals (LRT *p*-value < 0.05, posterior probability > 0.95), 21 of them were located in domains annotated by InterProScan (Supplementary Figs. 23, 24 and Supplementary Data 8), which may suggest they function in the conversion of loganic acid into secologanic acid in *C. acuminata*, although such an inference needs to be confirmed by further tests.

## Discussion

The availability of new biosynthetic enzymes or variants with desired catalytic abilities from diverse plants can assist us in metabolically engineering natural products. With the advent of

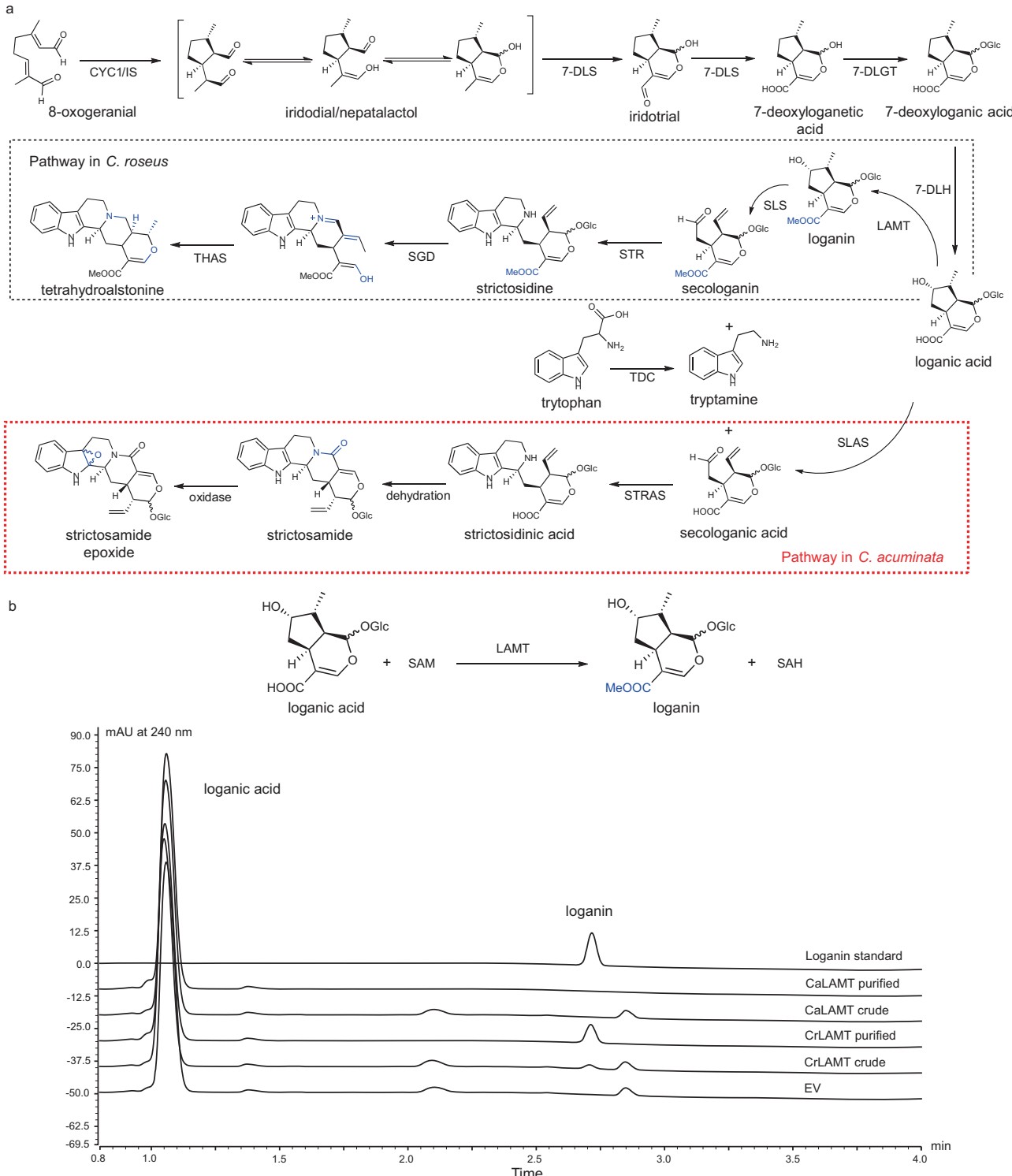

**Fig. 3 Comparison of camptothecin and vinblastine/vincristine biosynthesis pathways. a** Chemical representation of the iridoid pathway in *C. roseus* and *C. acuminata*. Different modifications resulting from reaction steps in the two species are indicated in blue. The camptothecin biosynthesis pathway in *C. acuminata* is outlined by the red-dotted box. **b** UPLC traces illustrate representative compound peaks for the target products of CaLAMT and CrLAMT.

new technologies and a dramatic decrease in price, genome sequencing has become a central and powerful tool for studies of plant metabolism in non-model species. A high-quality genome will serve as a foundation for discovering biosynthetic enzymes and gene clusters in such studies. Here, we report a high-quality chromosome-scale *C. acuminata* genome assembly produced using PacBio sequencing and Hi-C technology, which has

improved on the previously available genome assembly for this species in terms of both continuity and gene annotation. We discovered an independent WGD in *C. acuminata*. This WGD event and tandem duplication contributed greatly to the species-specific expansion of many *C. acuminata* gene families related to camptothecin biosynthesis, such as two *SLAS* genes, two *7-DLH-like* genes and other *CYP450* genes (Fig. 2a and Supplementary

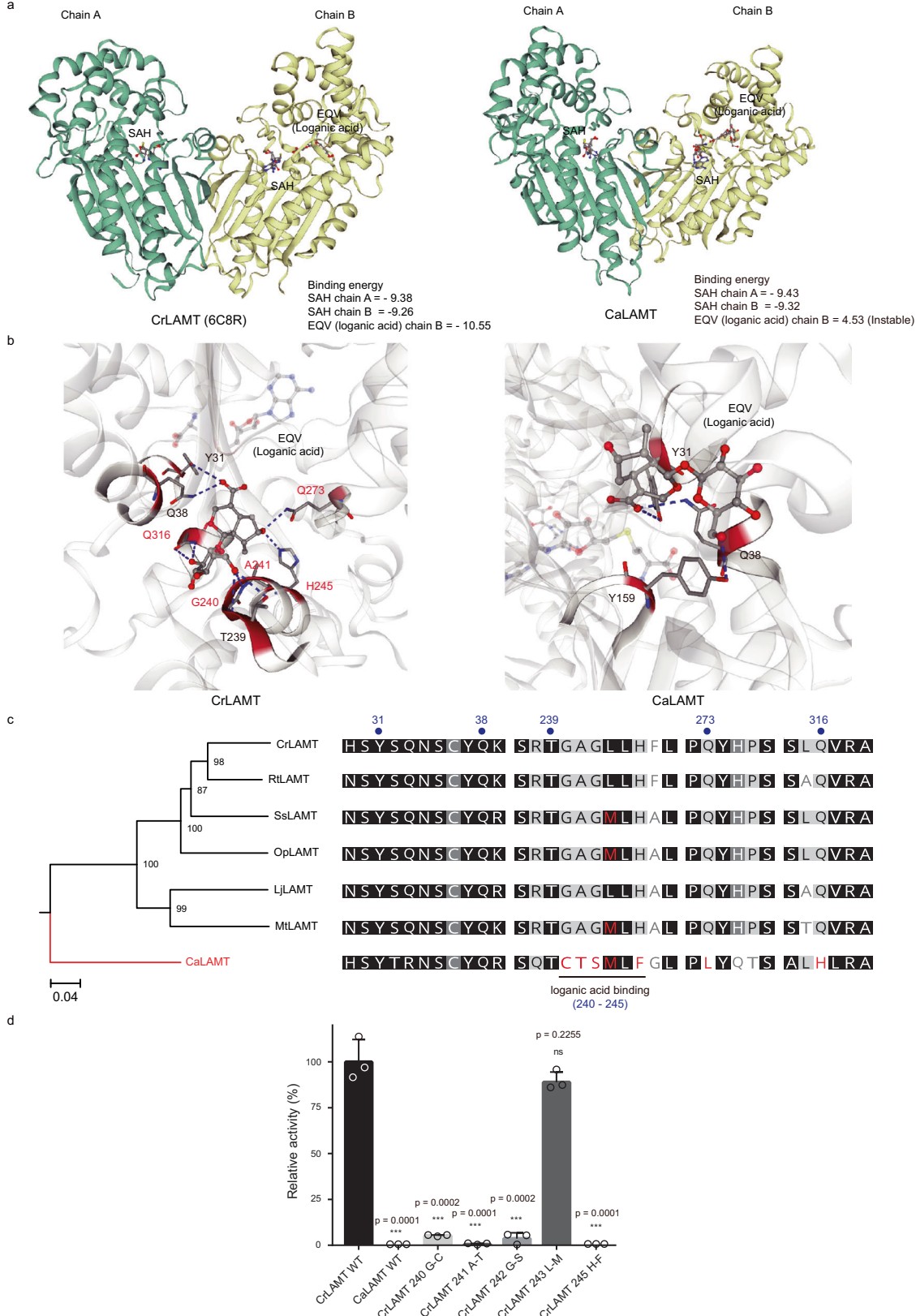

Figs. 15 and 22). We used this genome and gene annotation to identify genes related to camptothecin biosynthesis in *C. acuminata* and addressed how this pathway evolved.

The camptothecin biosynthetic pathway is similar to that for vinblastine/vincristine in *C. roseus* up to the production of loganic acid[11,18,19] (Fig. 3a). However, *C. roseus* uses LAMT and SLS to transform loganic acid into secologanin in two steps while *C. acuminata* converts loganic acid directly to secologanic acid using SLAS[11,36]. Based on the present reference genome, we used sequence analyses and functional tests to identify the genetic basis for this evolutionary divergence. Although numerous gene families related to terpene biosynthesis were expanded because of

**Fig. 4 Structural comparison of CrLAMT and CaLAMT and enzyme activity assay. a** Structures of CrLAMT and CaLAMT and their ligand binding energies calculated by AutoDock. **b** Docking results for loganic acid. Blue indicates hydrogen bonds while red indicates binding sites with mutations in CaLAMT compared with CrLAMT. **c** Phylogenetic relationship and protein sequence alignment of CrLAMT, CaLAMT, and LAMT homologs from other species. Binding sites and regions are shown in blue while sites with mutations are shown in red. RtLAMT: loganic acid *O*-methyltransferase from *Rauvolfia tetraphyla*; SsLAMT: loganic acid *O*-methyltransferase from *Strychnos spinosa*; OpLAMT: loganic acid *O*-methyltransferase from *Ophiorrhiza pumila*; LjLAMT: loganic acid *O*-methyltransferase from *Lonicera japonica*; MtLAMT: loganic acid *O*-methyltransferase from *Menyanthes trifoliata*. **d** Relative activities of CrLAMT WT, CaLAMT WT and the mutated CrLAMT compared to those found in CaLAMT. Relative enzyme activity was calculated using CrLAMT WT as a reference ($n = 3$ independent experiments for each enzyme). Significance was tested by a two-tailed unpaired *t*-test method (error bars, mean ± s.d) with asterisks indicating *p*-value (***$p < 0.001$; **$p < 0.01$; *$p < 0.05$). Source data underlying Fig. 4d are provided as a Source Data file.

the independent WGD event in *C. acuminata*, we found that there was only one candidate LAMT-encoding homolog in our genome. However, this gene showed low expression in all tissues and its product could not add a methyl to loganic acid as does the homolog in *C. roseus* (Figs. 3b, 4d, Supplementary Figs. 16–18, and Supplementary Data 7). We found that the function of CaLAMT had completely changed because of mutations in the ligand binding region compared with the LAMT homolog in *C. roseus* (Fig. 4 and Supplementary Fig. 18, and Supplementary Data 7). In addition, two *SLAS* genes derived from the recent WGD event in *C. acuminata* were highly expressed (Fig. 2a, Supplementary Fig. 22, and Supplementary Data 2) and both of their products could directly convert loganic acid into secologanic acid, although with differing efficiency[36]. They both differ from other homologs in closely related species with positive selection signals because of rapid evolution (Supplementary Figs. 23–24 and Supplementary Data 8). The functional divergence of the *LAMT* gene and positive evolution of the *SLAS* genes in *C. acuminata* may have eventually resulted in the evolution of an alternative MIA biosynthesis pathway for camptothecin. In addition, two SLASs (CYP72A565 and CYP72A610) have been reported to be able to convert 7-deoxyloganic acid into loganic acid, as 7-DLH does in *C. roseus*[36]. Based on our homology searches, functional annotation and phylogenetic analysis, we found that *CacGene10832* and *CacGene13171* encode candidate 7-DLH homologs in *C. acuminata* and both were highly expressed (Fig. 2a, c and Supplementary Data 2). These two gene copies similarly originated from the recent WGD event specific to *C. acuminata* in the same syntenic block as the two *SLAS* genes while the two *7-DLH-like* genes were tandemly repeated as was found for the two *SLAS* ones (Supplementary Fig. 22). It is necessary to test and compare the conversion efficiencies of these two *7-DLH-like* and two *SLAS* genes' products for different substrates and biosynthesis steps in the future in order to determine whether they underwent sub-functional divergence after the WGD and tandem duplication event.

It should be noted that camptothecin is also found in the remotely related *Ophiorrhiza pumila* (Rubiaceae)[37]. In this plant, LAMT, SLS and strictosidine synthase (STR) have the same functions as those of *C. roseus* and similarly produce loganin, secologanin and strictosidine[37] (Fig. 3a). All of the homologous and related genes originated from an independent whole-genome triplication[37]. In addition, strictosidine can be further converted into strictosamide in *O. pumila*, which leads to the production of camptothecin as in *C. acuminata* (Fig. 3a). However, *C. acuminata* synthesizes strictosidinic acid from secologanic acid and strictosamide in the pathway toward the final product camptothecin[11,37]. This alternative pathway for the production of camptothecin, which has fewer steps than that in *O. pumila*, may make a major contribution to the highly effective biosynthesis of camptothecin in *C. acuminata*. Furthermore, the parallel origins of camptothecin biosynthesis through different trajectories in these two distantly related groups highlight the diverse but convergent evolutionary histories of the WGD-derived genes, which

give rise to the same chemical. Artificial biosynthesis in the future could be designed based on the specialized pathway and associated candidate genes in *C. acuminata*.

## Methods

**DNA extraction and sequencing.** High-quality genomic DNA was extracted from fresh young leaves of *C. acuminata* using the cetyltrimethylammonium bromide (CTAB) method. The DNA was sheared, >30 Kb libraries were generated, and DNA was size-selected for inserts 20 Kb and sequenced using the PacBio Sequel platform for genome assembly. A total of six SMRT cells were sequenced. The PacBio subreads were obtained using the SMRT-link pipeline. In total 5.56 million subreads were generated with an N50 of 10.3 Kb and a mean length of 7.4 Kb (Supplementary Fig. 1). The target genome coverage of 100x was obtained with 41.35 Gb of sequencing data. An Illumina DNA-seq library was constructed using the same DNA. Paired-end 350 bp reads were sequenced on an Illumina HiSeq X Ten platform to produce a total of 47.86 Gb of data with genome coverage of 115x for error correction. For the Hi-C experiment, about 3 g of fresh young leaf tissue was ground to powder in liquid nitrogen. Then a Hi-C library was constructed by chromatin extraction and digestion, DNA ligation, purification and fragmentation. The resulting library was sequenced on an Illumina HiSeq X Ten platform in order to construct the chromosome-level genome assembly (Supplementary Table 1).

**Estimation of genome size.** The genome size of *C. acuminata* was estimated using the standard K-mer counting method. The term K-mer refers to a sequence with a length of kbp. We used clean Illumina short reads (size 17 bp) to calculate K-mer occurrence by means of Jellyfish[38]. The sequencing depth was estimated by determining the highest peak value in the frequency curve of K-mer occurrence distribution. The final genome size of *C. acuminata* estimated based on K-mer statistics was 412.44 Mb (Supplementary Fig. 2 and Supplementary Table 3).

**Genome assembly and chromosome construction.** Raw reads were error-corrected and assembled using the Falcon assembler with default parameters[39]. First, the corrected long reads were aligned with each other, according to overlap, to generate a string graph, thus forming the primary contig (p-Contig). Then we used FALCON-Unzip to find and classify heterozygous differences among them, integrated haplotype-fused contigs and re-assembled into haplotigs to obtain the updated primary contigs (p-Contigs) and haplotigs (h-Contigs). Finally, contigs were polished with Quiver using PacBio data and error-corrected by Pilon using 350 bp PE Illumina data to improve the accuracy of assembly[40,41], and ultimately obtain high-quality consensus sequences. In order to construct the chromosome-level genome, we employed LACHESIS software to cluster, reorder and orient the contig-scale genome assembly using Hi-C reads[42]. Then we manually checked the placement and orientation errors apparent in chromosomes using the Hi-C heat-map (Supplementary Fig. 3). The completeness of the genome assembly was assessed against a plant-specific database of 1440 single-copy orthologs by applying Benchmarking Universal Single-Copy Orthologs (BUSCO) with default settings[43].

**RNA-sequencing data collection and transcriptome assembly.** Fifteen developmental stage-specific RNA-sequencing (RNA-seq) datasets from *C. acuminata* were obtained from the NCBI SRA database (BioProject ID: PRJNA80029)[31] (Supplementary Table 19). We removed adapters and discarded reads with >10% *N* bases and reads having more than 20% low-quality bases (quality scores below 5) from these libraries using NGS QC Toolkit v 2.3.3[44]. Then we used nine paired-end sequencing reads, those from mature leaf, immature bark, entire root, young flower, immature fruit, mature fruit, cotyledons, upper stem and lower stem, to assemble a transcriptome with Trinity v 2.4.0[45] and generated a de novo 503 Mb assembly with a total of 589,795 transcripts and an N50 of 1,426 bp.

**Repeat annotation.** To structurally annotate repeat sequences in the *C. acuminata* genome, we began by discovering repetitive elements through application of RepeatModeler v 1.0.10 and RepeatMasker v 4.0.7[46,47]. RepeatModeler uses RECON and RepeatScout to predict interspersed repeats, then refines and classifies the consensus repeat models to build a repeat library. RepeatMasker was applied to

perform a homology-based repeat search throughout the *C. acuminata* genome using both the ab initio repeat database and Repbase. Finally, overlapping repeats belonging to the same repeat class were combined according to their coordinates in the genome. For overlapping repeats belonging to different repeat classes, the overlapped regions were split in the middle.

Long-terminal repeat retrotransposons (LTR-RTs) were initially identified using LTR Finder v 1.02 and LTRharvest[48,49]. LTR_retriever was then employed to filter out false LTR-RTs using three types of structural and sequence features: target site duplications, terminal motifs, and LTR-RT Pfam domains[50]. Finally, LTR-RTs were annotated by RepeatMasker using the non-redundant LTR-RT library constructed and the time of insertion of intact LTRs was provided by LTR_retriever.

**Gene prediction and function annotation**. For genome annotation by the PASA pipeline v 2.1.0[51] using the assembled transcriptome, we firstly applied end-trimming by seqclean to the transcriptome assembly. Then we ran PASA to align the transcript sequences to the genome assembly, and to predict ORFs and genes. In order to train the HMM model for Augustus v 3.2.2[52], we extracted complete, multi-exon genes, then removed redundant high identity genes (with an all-to-all identity cutoff of 70%), finally generating the best candidate and low identity gene models for training. In order to support genome annotation, we also used RNA-seq data aligned to the hard-masked genome assembly using HISAT2[53], then used bam2hints from Augustus to generate an intron hints file. Finally, we used this hints file to carry out gene prediction using Augustus. After prediction, we used PASA again to update the gff3 file for three rounds to add alternatively spliced isoforms to gene models.

Functional annotation was achieved by using NCBI BLAST + v 2.2.28[54] with cutoff e-values of 1e[-5] and max target sequences 20 to compare predicted proteins against public databases, including SwissProt and TrEMBL[55]. Best-hit BLAST results were then used to define gene functions. InterProScan-5.25-64.0[56] was employed to identify motifs and domains by matching against public databases. Gene Ontology identifiers for each gene were obtained using Blast2GO v 4.1[57] according to the blast results combined with InterPro GO entries. Existing Gene Ontology terms were then mapped to enzyme codes by Blast2GO and predicted proteins were submitted to KAAS to get KO numbers for KEGG pathway annotation[58].

**Gene family and phylogenetic analysis**. As references, protein sequences from seven species (*Coffea canephora*, *Daucus carota*, *Lactuca sativa*, *Catharanthus roseus*, *Rhododendron delavayi*, *Vitis vinifera* and *Arabidopsis thaliana*) were downloaded. For genes with alternative splicing variants, the longest transcript in each case was selected to represent the gene. Similarities between sequence pairs were calculated using blastp with cutoff e-values of 1e[−5]. Additionally, OrthoMCL v 2.0.9 was used with default parameters to identify gene family membership based on overall gene similarity combined with Markov Chain Clustering (MCL)[59].

Then we extracted single-copy orthologous genes from OrthoMCL results, protein sequences were aligned by MAFFT[60], Gblocks[61] was used to extract conserved sites from multiple sequence alignment results and a phylogenetic tree was constructed by RAxML[62] with the *A. thaliana* and *V. vinifera* datasets as the out-group; 1,000 bootstrap analyses were performed to test the robustness of each branch. In order to estimate species divergence time, a Bayesian relaxed molecular clock approach was used with MCMCTree in PAML[63] based on the calibration time for divergence between *A. thaliana* and *C. canephora* (110–124 Mya) obtained from the TimeTree database[64].

Gene families that had undergone expansion or contraction were identified in the eight sequenced species using CAFE[65]. CAFE parameters were set to *p*-value threshold = 0.05, and auto searching for the λ value. The algorithm in CAFE takes a matrix of gene family sizes in extant species as input and uses a probabilistic graphical model to ascertain the rate and direction of changes in gene family size across a given phylogenetic tree. Genes that belonged to specific expanded gene families were subjected to functional analysis using GO enrichment.

**Whole-genome duplication analysis and identification of tandemly repeated genes**. Homologous pairs of *C. acuminata* proteins were identified using an all-to-all search in blastp with an *e*-value cutoff of 1e[−9]. MCScanX[66] with default parameters was used to find collinear blocks, each containing at least five collinear gene pairs. Genes were further classified by duplicate gene-classifier in MCScanX. In order to look for whole-genome duplication (WGD) events, the downstream MCScanX script add_ka_and_ks_to_collinearity.pl was used to calculate the synonymous substitution rates per gene (Ks) between collinear genes in each pair out of *C. acuminata*, *R. delavayi* and *V. vinifera* and within each species. Whole-genome alignment of the *C. acuminata* and *V. vinifera* genomes was also carried out by LAST[67], and a dotplot was drawn to confirm the collinear relationship between these two genomes.

Identification of tandem repeat genes in the *C. acuminata* genome was based on three criteria: (1) two or more genes had >70% identity and 70% coverage according to blastp; (2) the pairwise gene distance was less than 100 kb; (3) there were no >10 genes lying between them on a single scaffold[26]. The genes so identified were subjected to functional analysis using GO enrichment.

**Gene expression and co-expression analysis**. We used HISAT2 to align RNA-seq short reads from 15 different tissues to the genome, with one mapped location being selected randomly for each read mapped to multiple locations. The expression level of each gene in terms of FPKM was computed by Cufflinks v 2.1.1[68]. A gene was considered to be expressed if its FPKM > 0. Differential gene expression analysis was conducted using the R package edgeR[69] with the parameter – dispersion = 0.1. For a gene to be considered to be differentially expressed it was required to have at least a twofold change in expression.

In order to find relationships between differentially expressed genes, we performed weighted gene co-expression analysis with the R package WGCNA[70]. The expression data were pre-filtered using the built-in quality control function. A signed co-expression network was constructed using a soft-thresholding power of 8 and default parameters. The only exception was the mergeCutHeight parameter, controlling the minimum distance between co-expression clusters, which was set to 0.25. We finally obtained 30 clusters for the genes. Then we used Cytoscape v 3.6.0[71] to display the network. Network statistics were calculated using NetworkAnalyzer in Cytoscape.

**Recombinant protein expression**. Coding sequences for the full-length loganic acid *O*-methyltransferases (LAMTs) from *C. roseus* and *C. acuminata* were initially reverse transcribed by PCR from leaf RNA (Supplementary Table 20) and the mutant sequences were synthesized by GeneArt (Thermo Fisher Scientific). The full-length cDNAs were cloned into the pESC-His expression vector with His tag using a ClonExpress II One Step Cloning Kit (Vazyme, China) (Supplementary Table 21 and Supplementary Fig. 25). *Saccharomyces cerevisiae* (WATII) was used as host yeast strain. Yeast transformation procedures were conducted with the lithium acetate method[72]. A single-colony grown on SD dropout medium containing 2% dextrose without histidine (Solarbio, China) was transferred into 15 mL SD dropout medium containing 2% dextrose. After incubating overnight at 30 °C on a rotary shaker set at 200 rpm, the culture was transferred into 200 mL of SD dropout medium containing 2% dextrose to obtain an OD600 of 0.4. All cells were then transferred into 200 mL SD dropout medium containing 2% galactose. The culture was incubated at 30 °C on a rotary shaker for a further 36 h. The cells were collected at $1,500 \times g$ for 5 min at 4 °C and washed with 2 mL sterile water.

The cells were crushed using acid-washed glass beads (425–600 μm, Sigma, USA) and the CrLAMT and CaLAMT proteins were purified with His-tag and an Ni-NTA spin column according to the instruction manual (Qiagen, USA). The protein was diluted in imidazole buffer using an ultrafiltration tube (Millipore). The crude and purified enzyme protein were stored at −20 °C until required for further enzyme activity testing.

**LAMT enzyme activity assay**. A total of 30 μL of crude or purified protein was mixed with 10 μL loganic acid (10 mM, final concentration 2 mM) and 10 μL *S*-adenosyl methionine (10 mM, final concentration 2 mM), and the mixture was incubated at 30 °C for 180 min. Then 400 μL methanol was added to stop the reaction, and after centrifugation at $20,000 \times g$ for 10 min, the supernatant was collected. The generation of loganin was measured on a Dionex UltiMate 3000 UHPLC system with Chromeleon software (Thermo Fisher Scientific, USA). Sample separation was performed using a Hypersil Gold C18 column ($100 \times 2.1$ mm, 1.9 μm; Thermo Fisher Scientific, USA), and the column temperature was set at 40 °C; the mobile phase was made up of 0.1% formic acid in water (A) and acetonitrile (B). The gradient was as follows: 10% B for 0.5 min; 10–40% B for 2 min; followed by re-equilibration of the column for 2.5 min with 10% B. The flow rate was set at 0.5 mL/min and injection volume was 2.0 μL. The sampler temperature was set at 10 °C. 240 nm was used as the detection wavelength.

The enzyme reaction mixture was also analyzed using an LC-ESI-MS/MS system (Nexera UHPLC LC-30A and AB SCIEX Triple Quad 5500 system). The reaction was stopped by chilling on ice instead of adding methanol. SPE was used for the removal of salt and protein. The sample was separated using a Shim-pack $2 \times 100$ column (SP XR-ODS $100 \times 2.0$ mm, Japan). The LC conditions were the same as stated above for UPLC. The ESI source operation parameters were as follows: ion source, turbo spray; source temperature (TEM) 500 °C; IonSpray voltage (IS) −4,500 V; Curtain gas (CUR), Ion source gas 1 (GS1) and Ion source gas 2 (GS2) were set at 40, 50, and 50 psi respectively; the Collision gas (CAD) was 9. For the MRM model, DP and CE for individual MRM transitions were done with optimization. For the MS2 model, the daughter ion of loganin (*m/z*, 389.2) was scanned for in the range 50–400. The protein concentration was measured using the A280 (nm) ultraviolet light absorption method. The relative enzyme activity was calculated using CrLAMT WT as a reference.

**Structural analyses and binding energy calculation**. Homology modeling was performed with SWISS-MODEL[73] using the closest template available (PDB ID: 6C8R). The results were inspected and rendered with PyMOL v 2.2.0[74]. Protein docking and binding energy calculation were done with AutoDock v 4.2.6 using local search parameters and default docking parameters[75].

**Selection analysis**. The cds fasta file of all sequences was aligned by MAFFT[60] and trimmed by Gblocks[61]. Then the Branch Site Model (BSM) and Branch Model

(BM) were applied to analyze alignments with codeml from the PAML v 4.9e package[63]. Chi2 in PAML v 4.9e package was used to calculate the LRT p-value.

**Reporting summary**. Further information on research design is available in the Nature Research Reporting Summary linked to this article.

## Data availability
Data supporting the findings of this work are available within the paper and its Supplementary Information files. A reporting summary for this article is available as a Supplementary Information file. The datasets and plant materials generated and analyzed during the current study are available from the corresponding author upon request. The PacBio long reads (SRR12042293) and Illumina short reads (SRR12042292 and SRR12042291) have been deposited in the NCBI Sequence Read Archive (SRA) database under BioProject PRJNA639006. The final chromosome-scale genome assembly [https://doi.org/10.6084/m9.figshare.12570599] and the GFF3 file [https://doi.org/10.6084/m9.figshare.12570614] are available in Figshare. The SwissProt and TrEMBL databases used in this study are available at https://www.uniprot.org. KEGG Pathway database is available at https://www.kegg.jp. Source data are provided with this paper.

## Code availability
Customized codes used in this study have been deposited in GitHub https://github.com/kmh1616/Cac_genome.

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

## Acknowledgements

This work was supported equally by the Strategic Priority Research Program of Chinese Academy of Sciences (XDB31010300), National Key Research and Development Program of China (2017YFC0505203), and the National Natural Science Foundation of China (grant numbers 31590821 and 91731301) and further by Fundamental Research Funds for the Central Universities SCU2019D013 and 2020SCUNL207, and National High-Level Talents Special Support Plan (10 Thousand of People Plan).

## Author contributions

J.L. designed the project. R.F., P.Z., and S.L. prepared materials and performed the experiments. M.K., X.Y., and Y.C. performed the bioinformatics analysis. M.K. wrote the manuscript. M.K., J.L., T.M., Y.Z., and X.X. revised the manuscript. All authors discussed the results and commented on the manuscript.

## Competing interests

The authors declare no competing interests.
