## [Peer Review File · Nature Communications]

REVIEWER COMMENTS

Reviewer #1 (Remarks to the Author):

NCOMMS-20-23704

This is, at first reading, a well-written, short and concise manuscript. It reports an improved genome sequencing, assembly and annotation of the medicinal plant *Camptotheca acuminata*. A lower quality sequencing and annotation of this plant genome was previously reported. The originality of the present manuscript, beyond improved quality of the genomic data, is the demonstration of a recent WGD in the *C. acuminata* genome.

Taking profit of genome annotation, the authors then tentatively identify the genes potentially contributing to the camptothecin (important anticancer drug) pathway in *C. acuminata*, based on phylogenetic analyses and clustering with the expected orthologs previously characterized in *Catharanthus roseus*. Interestingly, they point to the unexpected presence in *C. acuminata* of an ortholog of the of the gene encoding the methyltransferase converting loganic acid into loganin (LAMT) in *C. roseus*, and of two SLS-like (secologanin synthase) CYP72 genes. The latter three genes, based on the author's previous functional characterization of two 7-DLH-like multifunctional CYP72 enzymes and metabolic profiling of *C. acuminata* tissues, should not be not required for the camptothecin pathway.

In this part, the section from lines 208 to 217 is unclear. How are the upstream and downstream pathway segments distinguished or delineated in the analysis?

In an attempt to further reveal the potential complexity the pathway leading to camptothecin in *C. acuminata*, the expected LAMT ortholog is then expressed in yeast. The resulting enzyme is shown to be unable to catalyze the conversion of loganic acid into loganin. Low expression of this gene is detected in most parts of the plant. Protein modelling is in agreement with the absence of catalytic activity on loganic acid. The authors deduce that the loss of function of this LAMT is responsible for the divergence of the MIA pathway between *C. acuminata* and *C. roseus*. This is a plausible explanation, but does not constitute a demonstration that this loss of LAMT was the decisive event and not a consequence of CYP72 evolution. Novelty here mainly stems from a negative result.

Overall, the data and figures appear of good quality. However, conclusions are insufficiently supported and data more sound like the beginning of a story. Contrary to what might be expected from the misleading abstract (and discussion), the functional characterization and expression profiles of the two CYP72 genes catalyzing the formation of the loganic and secologanic acids and their expression have been previously reported. They are not SLS-like as mentioned in the manuscript, but 7-DLH like. The two SLS-like genes were not investigated previously, nor are they investigated in the current manuscript.

Many questions remain:

- The data suggest a pseudogenization of LAMT in *C. acuminata*, but do not test if it results from just a relaxation of selection pressure or positive selection after WGD (omega value?). Omega values on all sequence and active site might be compared.
- Can traces of a functional copy of LAMT be found in recently diverged genomes? (might be indicative of coexistence of two pathways at some stage).
- What was the cause of the loss of one LAMT copy and pseudogeneization of the second?
- What are the functions of the two SLS-like CYP72s? Are they both functional, expressed, under the same negative selection?
- Why are two copies of 7-DLH-like CYP72s and SLS-like CYP72s maintained?
- Has the acquisition of a dual function by *C. acuminata* CYP72s a role in the evolution of the pathway? (and do CYP72s from recently diverged plants have SLAS activity?)
- Does evolution of CYP72s to acquire dual function precede the loss of LAMT or the reverse? As both

the characterized 7-DLH-like enzymes show a dual activity and belong to two different linkage groups, this suggests that the dual function was acquired before WGD and maybe before LAMT loss of function.

At least some of these questions should be answered in the manuscript to provide a more solid evolutionary scenario.

Minor comments:

- I did not find the Supplemental Figure 19 in the reviewer's PDF.
- Some figures such as Figure 2C are too small.
- In Figure 3: the camptothecin pathway could be better highlighted.
- the sentence lines 264-265 is ambiguous and can be understood as *C. roseus* SLS can catalyze both 7-hydroxylation and ring opening, which is not the case. It has to be modified to avoid misunderstanding.

Reviewer #2 (Remarks to the Author):

The present study provides the results obtained from whole genome re-sequencing of *Camptotheca acuminata* with long reads from PacBio. The study corrects problems previously encountered in assembly and possible clustering of genes involved in assembly of natural products when using Illumina sequencing. The exploitation of PacBio is a highlight of this study to obtain further insights of the metabolic clusters occurring in medicinal plants.

The selection of genes for biochemical characterization focused on loganic acid O-methyltransferase from *C. acuminata*. Inspection of the previous Illumina *C. acuminata* database from October 2011 (Medicinal Plant Genomics) identified this gene (*caa_locus_129614_iso_1_len_1217_ver_4*).

How did the additional genome re-sequencing improve the prospects for characterizing this putative CaLAMT?

Was this because the original study did not produce a full length CaLAMT clone?

The biochemical characterization of the CaLAMT showed that it was not active with loganic acid as a substrate, compared with the activity of recombinant CrLAMT. The authors used this to provide supplementary evidence that loganic acid is converted to secologanic acid in *C. acuminata*. The authors make this a key component of their findings in the abstract of their article.

However, the enzymology is not well characterized. For example, the authors could have modified key amino acid residues on the CaLAMT to show how this catalytic activity could be re-established or how the CrLAMT activity could be lost. This would have given the possible evolutionary steps that might have occurred in the loss of this activity.

This was concept was illustrated when it was shown that *C. acuminata* plants accumulate camptothecin because of point mutations in DNA topoisomerase 1 that confers resistance to this alkaloid [Proc Nat Acad Sci (2008) 105: 6782-6786]. A camptothecin resistant DNA top1 was a necessary evolutionary step before these plants could accumulate this alkaloid.

Could it be that *C. acuminata* never evolved a functional LAMT? Instead they evolved a bifunctional deoxyloganic acid hydroxylase/secologanic acid synthase that would be responsible for providing the substrate for a putative strictosidinic acid synthase? The cloning and biochemical characterization of this bifunctional CYP is well-described in ACS Chem Biol 14:1091 that was published in 2019 (reference 32). While the authors refer to this study, a more details description of the biochemical properties of this bifunctional enzyme and its importance would have helped to highlight the role(s) of

the LAMT-like enzyme and the bifunctional CYP.

A neat experiment to do would have been to assay the bifunctional CYP together with the CrLAMT and appropriate co-substrates to see if Loganin and secologanin would be generated? This would have shown the importance of a loss of function LAMT or an never functional LAMT in the evolution of strictosidinic acid production.

Major issues

- The abstract makes unsubstantiated claims such as: "Camptotheca acuminata, a monoterpene indole alkaloid, is highly effective at curing diverse tumors." This statement should be much more explicit and careful in its claim.

- These broad statements are again repeated in the intro:

"It is the only natural plant active component that has been discovered so far to inhibit the action of topoisomerase I"

The references in support of this claim are from 1999 and 1985, respectively! If the authors are going to make such a bold claim that excludes the possibility that other plant-derived/inspired drugs target this enzyme, then they should have adequate, recent literature to back it up.

- CaLAMT is determined to have no function as compared to CrLAMT o An adequate explanation of the assay conditions needs to be given in Results and Methods

- o Results should clearly state whether the proteins were purified using a tag or if the activity was tested in vivo in bacteria or yeast

- o The methods section states that: "30ul crude or purified protein" was mixed with the substrate

- ♣ UThis is absolutely unacceptableU. If the activity of an enzyme is definitively ruled out the protein needs to be accurately quantified, and the same exact concentration of protein must be used in the positive control (CrLAMT) as in the test (CaLAMT) assays

- ♣ How much protein / what OD of microbial strains was used to determine activity? This needs to be clearly stated.

- ♣ Further, western blots should be included to confirm that the protein was expressed in both cases.

- o Loganic acid (substrate) was added at a concentration of 10 mM – this appears to be very high. Why was this concentration used? Is there no chance that the substrate could precipitate at this concentration?

- o Why is the substrate consumption so low for the active CrLAMT? ♣ It is possible based on this result (or the fact that the substrate concentration was so high) that the inactive CaLAMT also has activity, albeit at a lower level.

- ♣ The fact that neither the substrate nor the product is quantified despite having access to standards for both is confusing..

- It is strongly recommended that the authors repeat these assays and quantify both proteins and the substrate/product. Otherwise, the activity of CaLAMT cannot be ruled out.

- Page 15, Lines 263-266: "Further protein structures comparison and loganic acid-binding energies calculation of CrLAMT and CaLAMT also show that, although both have similar structures, site differences between them lead to changes in the ability to bind loganic acid: CaLAMT failed to bind this substrate stably or effectively." o This seems like a throwaway statement

- o Needs extra proof and additional explanation. What differences? What calculations?

- o This passage should be removed or explained further

- o The figure associated with this statement does not shed further insight

- It is stated that the two homologues of SLS, with additional SLAS activity, both can convert loganic acid to secologanic acid (in addition to the canonical loganin to secologanin), referencing Yang et al

- (ACS Chem Biol 2019). o Are there any differences in these two enzyme-encoding genes? Any activity differences? Any tissue expression pattern differences?
- o It seems that there are some aa changes between the previously reported genes (due to the re-sequencing efforts supposedly) – therefore, the enzyme assays should most likely be redone.
 - o The information on the two SLAS genes is lacking... this needs to be expanded.

Minor issues

- The authors should improve the clarity/grammar of the manuscript. In the abstract alone, there are several spelling and grammar errors: “leaded” instead of led; “converse” instead of convert.
- Why do the authors not follow conventional scientific nomenclature contractions of the genus, e.g., they say “*Cam. acuminata*” instead of *C. acuminata*?
- Line 46 of the intro, they refer to the plant as “*C. acuminata*” – the authors need to ensure that the species studied in the manuscript is correctly and consistently named...
- Page 5, line 89, “*ab initio*” should be italicized

Reviewer #3 (Remarks to the Author):

General comments:

This study presented an improved chromosome-level assembly of *Cam. acuminata*, and combined the published RNA-seq data, they authors identified an altered pathway of loganin/loganin acid that leads to the final production of camptothecin, an anti-tumor compound. The results derived from thorough bioinformatic analyses presented how more complete genome could facilitate gene mining and pathway discovery, which has both biological and medicinal significance in the plant genomics era, and highlighted the roles of genome duplication in reshaping the genome structure and genetic metabolic pathways. The author need to further highlight the novelties on methodology, data contribution and new knowledge in this study.

Minor comments:

1. L58. It would be clearer to state the version of the previously published *C. acuminata* genome here.
2. Providing additional details in Methods on the comparison of the two genome assembly versions would be very helpful, e.g., demonstrating how the new assembly could better facilitate gene discovery.
3. L110. Is the significance supported by any statistical test?
4. L120. None of the enriched functions of *Cam. acuminata*-specific genes were involved in the biosynthesis of indole. I wonder whether SLAS is unique to *Cam. acuminata*? Or both *Cam. acuminata* and *Cat. roseus* maintains this gene. If they do, what’s the difference between them? Why it does not convert loganic acid to secologanic acid directly?
5. L154. It is better to and specify and quantify details to support the conclusion that “The *Cam. acuminata* specific WGD and tandem duplication were the key contributor to gene family expansions in this species”.
6. L157-160. It should be careful to avoid any over-interpretation on the results of gene expressions when collinear pairs were compared. Only one of the multiple hits were randomly selected for the measurement of gene expression might introduce errors in this analysis.
7. L163. How do we know it is the duplicated one but not the original/ancient copy that underwent functional diversification? Any syntenic block support that?
8. It is a nice work of the authors to have identified the genes involved in the altered pathway leading to the biosynthesis of camptothecin. However, I don’t think this work highlighted the advantage of genome in the mining of new genes, compared to transcriptome, as the authors claimed. I noticed that those genes that the authors have identified were highly expressed, and “the high content of camptothecin in *Cam. acuminata* tissues is likely to be attributable to the constant and high level of expression of these genes” (L204), so these genes are easy to detect in transcriptomic data. I assume people will have similar findings if they were focusing on the same questions, regardless of the data

sources.

9. L344. I didn't see any description on genome size estimation in the main text. Remove this paragraph if it is not involved in the work. Please check.

10. Some wordings in the text could be more concise. For example, L12 "catalyze the production of loganin by loganic acid" needs rephrase. L13 "as is the case in" could be better stated as "contrary to the case in".

REVIEWER COMMENTS

Reviewer #1 (Remarks to the Author):

NCOMMS-20-23704

This is, at first reading, a well-written, short and concise manuscript. It reports an improved genome sequencing, assembly and annotation of the medicinal plant *Camptotheca acuminata*. A lower quality sequencing and annotation of this plant genome was previously reported. The originality of the present manuscript, beyond improved quality of the genomic data, is the demonstration of a recent WGD in the *C. acuminata* genome.

Taking profit of genome annotation, the authors then tentatively identify the genes potentially contributing to the camptothecin (important anticancer drug) pathway in *C. acuminata*, based on phylogenetic analyses and clustering with the expected orthologs previously characterized in *Catharanthus roseus*. Interestingly, they point to the unexpected presence in *C. acuminata* of an ortholog of the of the gene encoding the methyltransferase converting loganic acid into loganin (LAMT) in *C. roseus*, and of two SLS-like (secologanin synthase) CYP72 genes. The latter three genes, based on the author's previous functional characterization of two 7-DLH-like multifunctional CYP72 enzymes and metabolic profiling of *C. acuminata* tissues, should not be not required for the camptothecin pathway.

In this part, the section from lines 208 to 217 is unclear. How are the upstream and downstream pathway segments distinguished or delineated in the analysis?

Response: We added some description for putative biosynthetic pathway construction in the revised version. We outlined the putative biosynthetic camptothecin pathway based on the KEGG database, previously published results (Sadre R, *et al.*, 2016), expression profile of each candidate gene in 15 tissues and co-expression modules. The enzymes and upstream and downstream relationships of each step can be clearly found in Fig. 2A. The important gene copies from the seco-iridoid pathway were grouped together in the same WGCNA module, and the related steps are continuous.

In an attempt to further reveal the potential complexity the pathway leading to camptothecin in *C. acuminata*, the expected LAMT ortholog is then expressed in yeast. The resulting enzyme is shown to be unable to catalyze the conversion of loganic acid into loganin. Low expression of this gene is detected in most parts of the plant. Protein modelling is in agreement with the absence of catalytic activity on loganic acid. The authors deduce that the loss of function of this LAMT is responsible for the divergence of the MIA pathway between *C. acuminata* and *C. roseus*. This is a plausible explanation, but does not constitute a demonstration that this loss of LAMT was the decisive event and not a consequence of CYP72 evolution. Novelty here mainly stems from a negative result.

Response: We spent 5 months to establish a recombinant protein expression system based on site mutation and do enzyme activity assay. We demonstrated that most of the mutations in binding region of the LAMT gene in *C. acuminata* comparing with *Catharanthus roseus*, decreased greatly or abolished the enzyme activity. In addition, our results showed that two SLAS genes in *C. acuminata* experienced positive evolution comparing with SLS-like genes of other close related species, which may provide genetic bases for them to convert loganic acid into secologanic acid instead.

Overall, the data and figures appear of good quality. However, conclusions are insufficiently supported and data more sound like the beginning of a story. Contrary to what might be expected from the misleading abstract (and discussion), the functional characterization and expression profiles of the two CYP72 genes catalyzing the formation of the loganic and secologanic acids and their expression have been previously reported. They are not SLS-like as mentioned in the manuscript, but 7-DLH like. The two SLS-like genes were not investigated previously, nor are they investigated in the current manuscript.

Response: Although the two SLAS genes could also convert 7-deoxyloganic acid into loganic acid like 7-DLH genes did in *C. acuminata* based on the previous research (Yang Y, *et al.*, 2019), phylogenetic analysis in both our result (Fig. 2C) and the previous research (Yang Y, *et al.*, 2019) showed that these two genes are clustered in the

corresponding branch of the SLS gene of *Catharanthus roseus*, instead of 7-DLH. Based on our genome sequence, gene annotation and phylogenetic analysis, we found that CacGene10832 and CacGene13171 are obvious 7-DLH homologs in *C. acuminata* and both are highly expressed (Figure 2A and 2C, Supplemental Table 19). These two 7-DLH-like genes similarly originated from the recent WGD event specific to *C. acuminata* (Supplemental Figure 20). We also tried many times to test and compare converting efficiencies of these four genes in the past half year. However, we failed to extract 7-dexoxyloganic acid and secologanic acid (as reference to examine converting consequences) due to the lack of the fresh materials because of the COVID-19 pandemic, and they also could not be bought in the market. We acknowledge and discuss this caveat in the discussion section. We believe that two SLAS genes had some redundant functions with two 7-DLH-like genes. However, failure to test functions of these two 7-DLH-like genes in *C. acuminata* will not affect our total story and conclusion: in *C. acuminata*, degeneration of the LAMT gene and positive evolution of two SLAS genes together evolved a new alternative MIA pathway, which finally results in production of camptothecin in *C. acuminata*.

Many questions remain:

- The data suggest a pseudogenization of LAMT in *C. acuminata*, but do not test if it results from just a relaxation of selection pressure or positive selection after WGD (omega value?). Omega values on all sequence and active site might be compared.

Response: As suggested by the referee, we added the selection analysis using branch-site model in paml v 4.9e setting CaLAMT as the foreground branch, but the result shows that CaLAMT did not have sites under significant positive selection (LRT p value < 0.05, posterior probability > 0.95) (Supplemental Table 23). Most sites of the clean alignment (255 / 300) were calculated with a ω (dN/dS) value > 10 and the estimates of parameters ω_1 of site class 2a and 2b were 999, which were untrustworthy. We thought that such results might arise from the lack of enough published LAMT sequences from the closely

related species and therefore too large sequence differences between the used LAMT sequences used here.

- Can traces of a functional copy of LAMT be found in recently diverged genomes? (might be indicative of coexistence of two pathways at some stage).

Response: The closest functional copy published in recently diverged genomes is the CrLAMT in *C. roseus* based on the phylogenetic tree in Fig. 1A. CaLAMT has 8 mutate amino acid sites in binding regions or sites compared with CrLAMT, and 7 of them can greatly reduce or abolish the enzyme activity. We added the new experimental results in the revise manuscript.

- What was the cause of the loss of one LAMT copy and pseudogeneization of the second?

Response: Because of lack of enough published LAMT sequences of the closely related species, it is difficult to trace this process as stated before. However, our new experiments obviously suggested that mutations in CaLAMT lead to its functional loss. The positive evolution of two SLAS genes may also account for such a loss.

- What are the functions of the two SLS-like CYP72s? Are they both functional, expressed, under the same negative selection?

Response: The two SLS-like CYP72s (called as two SLAS genes here) are both functional, and expressed, but their expression level and enzyme activities have some difference, which can be found in Fig 2A and Yang et al., 2019. Maybe these two genes have experienced sub-functional divergences at different developments and times after WGD event (Supplemental Fig. 20), which enforce their converting functions. We added this part in the revised manuscript.

- Why are two copies of 7-DLH-like CYP72s and SLS-like CYP72s maintained?

Response: Similarly, both 7-DLH-like CYP72s may have sub-functional divergences at different developments and times after WGD event (Supplemental Fig. 20). In addition, both SLS-like CYP72s (two SLAS genes) could also convert 7-deoxyloganic acid into loganic acid mentioned above in the previous study (Yang Y, *et al.*, 2019) like 7-DLH. We suggested that the newly found high-expressed 7-DLH-like CYP72s and two SLAS genes evolved by WGD might have different reaction efficiencies in performing different converting steps. This may require further functional verification.

- Has the acquisition of a dual function by *C. acuminata* CYP72s a role in the evolution of the pathway? (and do CYP72s from recently diverged plants have SLAS activity?)

Response: No enzyme with SLAS activity had been reported in other species except for *C. acuminata* so far. Since we do not have the reference substance (pure secologanic acid extracted from this or other species) as stated before, we cannot verify it by ourselves. But based on our evolutionary analysis for two SLAS copies of *C. acuminata* and SLS-like genes in other species, these two SLAS copies in *C. acuminata* are under positive selection as mentioned in the revised manuscript. This positive evolution may lead to the current new function of two SLAS genes.

- Does evolution of CYP72s to acquire dual function precede the loss of LAMT or the reverse? As both the characterized 7-DLH-like enzymes show a dual activity and belong to two different linkage groups, this suggests that the dual function was acquired before WGD and maybe before LAMT loss of function.

Response: This conclusion has to be reasoned from extensive comparisons between all LAMT sequences of the closely related species. However, up to now, few related species have genomes, which restricts our comparisons. Another way refers to experimental test of reaction efficiencies of these genes with mutate sites. As suggested before, some the

reference substances could not be bought in the market (we also asked the labs who published related papers and all answered that they had used up the purified metabolites). We acknowledge and discuss this caveat in the discussion section. However, all of these caveats do not affect our major conclusions.

At least some of these questions should be answered in the manuscript to provide a more solid evolutionary scenario.

Response: Thank you for your kindness and understanding. We established a recombinant protein expression system based on site mutation in CaLAMT and did enzyme activity assays. We demonstrated that most of the mutations in binding region of the LAMT gene in *C. acuminata* comparing with *Catharanthus roseus* decreased greatly or abolished the enzyme activity. In addition, we demonstrated the positive evolutions of two SLAS genes comparing with the homologs of other species. These two lines of evidence greatly strengthen the final conclusion although we could not address all questions.

Minor comments:

- I did not find the Supplemental Figure 19 in the reviewer's PDF.

Response: We can clearly find the Supplemental Figure 19 in the PDF version on the website. It is now Supplemental Figure 20 in the revised manuscript.

- Some figures such as Figure 2C are too small.

Response: Thanks for your suggestion. We have modified figures to be as large as possible.

- In Figure 3: the camptothecin pathway could be better highlighted.

Response: Thanks for your suggestion. We highlighted the camptothecin pathway in Figure 3.

- the sentence lines 264-265 is ambiguous and can be understood as *C. roseus* SLS can catalyze both 7-hydroxylation and ring opening, which is not the case. It has to be modified to avoid misunderstanding.

Response: Thanks for your suggestion. We have modified this sentence.

Reviewer #2 (Remarks to the Author):

The present study provides the results obtained from whole genome re-sequencing of *Camptotheca acuminata* with long reads from PacBio. The study corrects problems previously encountered in assembly and possible clustering of genes involved in assembly of natural products when using Illumina sequencing. The exploitation of PacBio is a highlight of this study to obtain further insights of the metabolic clusters occurring in medicinal plants.

The selection of genes for biochemical characterization focused on loganic acid O-methyltransferase from *C. acuminata*. Inspection of the previous Illumina *C. acuminata* database from October 2011 (Medicinal Plant Genomics) identified this gene (`caa_locus_129614_iso_1_len_1217_ver_4`).

How did the additional genome re-sequencing improve the prospects for characterizing this putative CaLAMT?

Was this because the original study did not produce a full length CaLAMT clone?

Response: Yes. The previous published CaLAMT clone (`caa_locus_129614_iso_1_len_1217_ver_4`) was complete. However, the previous studies failed to make a conclusion about functional loss of CaLAMT possibly because they assumed that this gene is still workable or there might have another workable copy. It should be noted that our

reference genome (Complete BUCSO: 1,270 / 1,440 94.9%) has a broadly more complete genes set than transcriptome and the previously published genome (Supplemental Table 16). Therefore, our genome can replenish the discovery of more candidate gene copies and their gene families involved in camptothecin biosynthesis. In addition, this chromosome-level genome assembly makes it possible to study the relative position of genes on chromosomes and their collinearity, especially the positional relationship among the two 7-DLH-like copies and two SLAS copies displayed in Supplemental Fig. 20.

The biochemical characterization of the CaLAMT showed that it was not active with loganic acid as a substrate, compared with the activity of recombinant CrLAMT. The authors used this to provide supplementary evidence that loganic acid is converted to secologanic acid in *C. acuminata*. The authors make this a key component of their findings in the abstract of their article.

However, the enzymology is not well characterized. For example, the authors could have modified key amino acid residues on the CaLAMT to show how this catalytic activity could be re-established or how the CrLAMT activity could be lost. This would have given the possible evolutionary steps that might have occurred in the loss of this activity.

Response: Thanks for your suggestion. In the revised manuscript, we had modified key mutate sites and examined these mutations affect LAMT activities. The relative enzyme activities were calculated using CrLAMT (KF415116) WT as a reference. In fact, each of 7 sites among 8 mutations in CaLAMT could greatly reduce or totally abolish the enzyme activity (Fig 4C, Supplemental Fig 18 and Supplemental Table 24).

This was concept was illustrated when it was shown that *C. acuminata* plants accumulate camptothecin because of point mutations in DNA topoisomerase 1 that confers resistance to this alkaloid [Proc Nat Acad Sci (2008) 105: 6782-6786]. A camptothecin resistant DNA top1

was a necessary evolutionary step before these plants could accumulate this alkaloid.

Could it be that *C. acuminata* never evolved a functional LAMT? Instead they evolved a bifunctional deoxyloganic acid hydroxylase/secologanic acid synthase that would be responsible for providing the substrate for a putative strictosidinic acid synthase? The cloning and biochemical characterization of this bifunctional CYP is well-described in ACS Chem Biol 14:1091 that was published in 2019 (reference 32). While the authors refer to this study, a more details description of the biochemical properties of this bifunctional enzyme and its importance would have helped to highlight the role(s) of the LAMT-like enzyme and the bifunctional CYP.

Response: Thanks for your suggestion. We added more description about the LAMT-like enzyme and the bifunctional CYPs. *C. acuminata* might have never evolved a functional LAMT but likely evolved a bifunctional deoxyloganic acid hydroxylase/secologanic acid synthase. This is possible and can account for the inactivation or functional alteration of the LAMT ortholog. However, positive evolutions of two SLAS genes may comprise another reason why *C. acuminata* produces different indole alkaloids by using a similar pathway with *C. roseus* (Supplemental Fig. 21 and Supplemental Table 25). We added all related discussion in the revised manuscript. However, it is difficult to test CaLAMT had never been functional since its origin because the homologous sequences of the closely related species were not enough for such a comparison. Our current evidence through point-mutation experiments demonstrated that each of 7 sites among 8 mutations in CaLAMT (comparing with CrLAMT) greatly reduced or totally abolished enzyme activity. However, we do not know how these mutations developed. These caveats do not affect our major conclusion that functional loss of CaLAMT and positive evolutions of two SLAS genes had led to the origin of the camptothecin biosynthesis pathway.

A neat experiment to do would have been to assay the bifunctional CYP together with the CrLAMT and appropriate co-substrates to see if Loganin and secologanin would be generated?

This would have shown the importance of a loss of function LAMT or an never functional LAMT in the evolution of strictosidinic acid production.

Response: The two SLAS genes could convert loganin into secologanin, which had already been confirmed recently by Yang Y et al. (2019). However, there was no loganin in *C. acuminata* itself, but secologanic acid was isolated and present instead (Sadre R, et al., 2016).

Major issues

- The abstract makes unsubstantiated claims such as: “Camptotheca acuminata, a monoterpene indole alkaloid, is highly effective at curing diverse tumors.” This statement should be much more explicit and careful in its claim.
- These broad statements are again repeated in the intro:

“It is the only natural plant active component that has been discovered so far to inhibit the action of topoisomerase I”

The references in support of this claim are from 1999 and 1985, respectively! If the authors are going to make such a bold claim that excludes the possibility that other plant-derived/inspired drugs target this enzyme, then they should have adequate, recent literature to back it up.

Response: Thanks for your suggestion. We have changed our statement and added the recent references.

- CaLAMT is determined to have no function as compared to CrLAMT o An adequate explanation of the assay conditions needs to be given in Results and Methods
o Results should clearly state whether the proteins were purified using a tag or if the activity was tested in vivo in bacteria or yeast.

o The methods section states that: “30ul crude or purified protein” was mixed with the substrate. This is absolutely unacceptable. If the activity of an enzyme is definitively ruled out the protein needs to be accurately quantified, and the same exact concentration of protein must be used in the positive control (CrLAMT) as in the test (CaLAMT) assays

How much protein / what OD of microbial strains was used to determine activity? This needs to be clearly stated. Further, western blots should be included to confirm that the protein was expressed in both cases.

Response: We added the details in the Method section. The full-length cDNAs were cloned into the pESC-His expression vector with His tag using a ClonExpress II One Step Cloning Kit (Vazyme, China). We purified the protein using His-tag and an Ni-NTA spin column according to the instruction manual (Qiagen, USA). In the further WT and mutation activity comparison added in the revised manuscript, we used A280 (nm) ultraviolet light absorption method to measure the protein concentration and calculate the standardized relative enzyme activity using CrLAMT (KF415116) WT as a reference (Fig 4C, Supplemental Fig 18 and Supplemental Table 24).

o Loganic acid (substrate) was added at a concentration of 10 mM – this appears to be very high. Why was this concentration used? Is there no chance that the substrate could precipitate at this concentration? Why is the substrate consumption so low for the active CrLAMT? □□It is possible based on this result (or the fact that the substrate concentration was so high) that the inactive CaLAMT also has activity, albeit at a lower level. The fact that neither the substrate nor the product is quantified despite having access to standards for both is confusing...

It is strongly recommended that the authors repeat these assays and quantify both proteins and the substrate/product. Otherwise, the activity of CaLAMT cannot be ruled out.

Response: Thank you for excellent suggestions. We added all details in the revised manuscript. 10 mM is the concentration of 10μL and the final concentration is 2mM. In the further WT and mutation activity comparison, we used A280 (nm) ultraviolet light

absorption method to measure the protein concentration and calculate the standardized relative enzyme activity using CrLAMT (KF415116) WT as a reference. The experiments were repeated three times (Fig 4C, Supplemental Fig 18 and Supplemental Table 24).

- Page 15, Lines 263-266: “Further protein structures comparison and loganic acid-binding energies calculation of CrLAMT and CaLAMT also show that, although both have similar structures, site differences between them lead to changes in the ability to bind loganic acid: CaLAMT failed to bind this substrate stably or effectively.”
 - o This seems like a throwaway statement
 - o Needs extra proof and additional explanation. What differences? What calculations?
 - o This passage should be removed or explained further
 - o The figure associated with this statement does not shed further insight

Response: Thanks for your suggestion. We added the related details.

- It is stated that the two homologues of SLS, with additional SLAS activity, both can convert loganic acid to secologanic acid (in addition to the canonical loganin to secologanin), referencing Yang et al (ACS Chem Biol 2019).
 - o Are there any differences in these two enzyme-encoding genes? Any activity differences? Any tissue expression pattern differences?
 - o It seems that there are some aa changes between the previously reported genes (due to the re-sequencing efforts supposedly) – therefore, the enzyme assays should most likely be redone.
 - o The information on the two SLAS genes is lacking... this needs to be expanded.

Response: Thanks for your suggestion. Both genes are highly expressed in all tissues while CacGene10833 (CYP72A565) has higher expression level than CacGene13172 (CYP72A610) (Fig. 2A, Supplemental Table 19). CYP72A565 also has a higher enzyme activity than CYP72A610 (Yang et al. 2019). We expanded the information about these

two SLAS genes in the revised manuscript. CacGene10833 and CYP72A565 do not have any aa changes while CacGene13172 and CYP72A610 which was reported before do have. This difference may be caused by SNPs among different individuals. As mentioned before, we could not examine their activity differences due to the lack of the purified secologanic acid.

Minor issues

- The authors should improve the clarity/grammar of the manuscript. In the abstract alone, there are several spelling and grammar errors: “leaded” instead of led; “converse” instead of convert.

Response: Thanks for your suggestion. We have tried our best to polish the language in the revised manuscript.

- Why do the authors not follow conventional scientific nomenclature contractions of the genus, e.g., they say “*Cam. acuminata*” instead of *C. acuminata*?

Response: As suggested by the reviewer, we have changed all the ‘*Cam. acuminata*’ into ‘*C. acuminata*’ and ‘*Cat. roseus*’ into ‘*C. roseus*’.

- Line 46 of the intro, they refer to the plant as “*C. accuminata*” – the authors need to ensure that the species studied in the manuscript is correctly and consistently named...
- Page 5, line 89, “ab initio” should be italicized

Response: Thank the reviewer for careful reading. We have corrected these words in our revised manuscript.

Reviewer #3 (Remarks to the Author):

General comments:

This study presented an improved chromosome-level assembly of *Cam. acuminata*, and combined the published RNA-seq data, they authors identified an altered pathway of loganin/loganin acid that leads to the final production of camptothecin, an anti-tumor compound. The results derived from thorough bioinformatic analyses presented how more complete genome could facilitate gene mining and pathway discovery, which has both biological and medicinal significance in the plant genomics era, and highlighted the roles of genome duplication in reshaping the genome structure and genetic metabolic pathways. The author need to further highlight the novelties on methodology, data contribution and new knowledge in this study.

Minor comments:

1. L58. It would be clearer to state the version of the previously published *C. acuminata* genome here.

Response: Thanks for your suggestion. We added and compared the previously published *C. acuminata* genome and our newly assembled genome here.

2. Providing additional details in Methods on the comparison of the two genome assembly versions would be very helpful, e.g., demonstrating how the new assembly could better facilitate gene discovery.

Response: We demonstrated how the new assembly could better facilitate gene discovery as we displayed in Supplemental Fig. 5-7. The new assembly has higher gene annotations. The chromosome-level genome assembly makes it possible to study the relative position of genes on chromosomes and their collinearity. For example, the positional relationships among the two 7-DLH-like copies and two SLAS copies could be further displayed (Supplemental Fig. 19). We added more descriptions in the revised manuscript.

3. L110. Is the significance supported by any statistical test?

Response: Yes, significance was tested by Wilcoxon method with p-value < 2.2e-16. We add it in the main text.

4. L120. None of the enriched functions of *Cam. acuminata*-specific genes were involved in the biosynthesis of indole. I wonder whether SLAS is unique to *Cam. acuminata*? Or both *Cam. acuminata* and *Cat. roseus* maintains this gene. If they do, what's the difference between them? Why it does not convert loganic acid to secologanic acid directly?

Response: As we mentioned in the revised manuscript, *C. roseus* have two SLS genes, and two SLAS genes found here were clustered with these two SLS genes (which were therefore called as SLS-like genes in *C. acuminata* in the previous study, Yang Y, *et al.*, 2019). But two SLS genes from *C. roseus* could only convert loganin into secologanin while two SLAS genes of *C. acuminata* could not only covert loganin into secologanin, but also loganic acid into secologanic acid (Yang Y, *et al.*, 2019). Therefore, the positive evolutions of two SLAS genes might account for their new functions to convert loganic acid into secologanic acid. We also expanded this information in the revised manuscript.

5. L154. It is better to and specify and quantify details to support the conclusion that “The *Cam. acuminata* specific WGD and tandem duplication were the key contributor to gene family expansions in this species”.

Response: Thanks for your suggestion. We added the percentage of WGD and tandem repeat genes in expanded gene families in the revised manuscript (Supplemental Table 13).

6. L157-160. It should be careful to avoid any over-interpretation on the results of gene expressions when collinear pairs were compared. Only one of the multiple hits were randomly

selected for the measurement of gene expression might introduce errors in this analysis.

Response: We agree with the reviewer's assessment. We deleted this comparison of collinear pairs' gene expression level to avoid the errors. This part has no direct connection with the main context and conclusion.

7. L163. How do we know it is the duplicated one but not the original/ancient copy that underwent functional diversification? Any syntenic block support that?

Response: We had hoped to express that 'the differential expression of collinear gene pairs might indicate that one of the two copies probably undergone functional diversification by gaining novel responses to differing environmental conditions at the expression level'. However, because of our careless description, this sentence causes misunderstanding. Now we deleted this part.

8. It is a nice work of the authors to have identified the genes involved in the altered pathway leading to the biosynthesis of camptothecin. However, I don't think this work highlighted the advantage of genome in the mining of new genes, compared to transcriptome, as the authors claimed. I noticed that those genes that the authors have identified were highly expressed, and "the high content of camptothecin in *Cam. acuminata* tissues is likely to be attributable to the constant and high level of expression of these genes" (L204), so these genes are easy to detect in transcriptomic data. I assume people will have similar findings if they were focusing on the same questions, regardless of the data sources.

Response: We admit that some genes can be found in transcriptome data. However, the high-quality reference genome has an obviously more complete gene sets than transcriptome (Supplemental Table 16), which can replenish the discovery of more candidate gene copies and their gene families involved in camptothecin biosynthesis. The chromosome-level genome assembly makes it possible to study relative positions of genes on chromosomes and their collinearity. For example, the positional relationship among

the two 7-DLH-like copies and two SLAS copies could be further displayed (Supplemental Fig. 19).

9. L344. I didn't see any description on genome size estimation in the main text. Remove this paragraph if it is not involved in the work. Please check.

Response: We sincerely thank the reviewer for careful reading. We added the description related to genome size estimation in the main text and not just in the Method part.

10. Some wordings in the text could be more concise. For example, L12 "catalyze the production of loganin by loganic acid" needs rephrase. L13 "as is the case in" could be better stated as "contrary to the case in".

Response: Thanks for your suggestion. We have modified these sentences.

Sadre R, *et al.* Metabolite diversity in alkaloid biosynthesis: a multilane (diastereomer) highway for camptothecin synthesis in *Camptotheca acuminata*. *The Plant Cell* **28**, 1926-1944 (2016).

Yang Y, *et al.* Bifunctional Cytochrome P450 Enzymes Involved in Camptothecin Biosynthesis. *ACS Chem Biol* **14**, 1091-1096 (2019).

REVIEWER COMMENTS

Reviewer #1 (Remarks to the Author):

NCOMMS-20-23704A

The reviewers deployed some efforts to improve the manuscript. The first part, dedicated to the *C. acuminata* genome resequencing, reads fine and seems coherent. Unfortunately the second, dedicated to the evolution of the camptothecin pathway, is still inadequate, both from a redactional and conceptual points of view.

The manuscript text is awkward and inappropriate, with meaningless sentences, ill-used English, repetitions, many mistakes that are not typos since repeated, even the corrections kindly suggested in the first round of review by Reviewer 2 have not been implemented (and there are too many to list). Gene names are not italicized, and there is a big mix-up between genes and proteins, resulting in genes with enzymatic activities. Even gene expression is mentioned as having enzymatic activity in the conclusion. When genes do not encode proteins having the same activity, they should be called homologs and not orthologs, as is the case in the manuscript. P450 families are not defined based on their activity (as mentioned in text) but on their phylogeny and based on decision of nomenclature committee (nevertheless the family labelling in sup figure 15 seems consistent and most likely appropriate). In many places the text should be shortened to remove redundant and verbose sections. What stems from previous work should be more clearly stated and not mixed up with new data provided in the manuscript to introduce ambiguity. Note also that at least some reference numbering does not seem to be correct. The authors should ask for the assistance of an experienced researcher to update their final version.

The story itself, based on the interesting statement that different pathways evolved in different species to generate the same or related compounds, is interesting, but it is not correctly exploited.

The demonstration that the CaLAMT homolog has no LAMT activity is convincing.

Though :

- 1) According to the information provided, all the residues differing between CrLAMT and CaLAMT-like proteins are located around the active site and all, except one, lead to the loss of LAMT activity. This is extremely unlikely to occur when a gene undergoes pseudogeneization, whereupon mutations should be dispatched on the whole sequence. So, the data provided rather hint at acquisition of new function.
- 2) In the figure 3A, showing the models of the whole CrLAMT and CaLAMT proteins does not provide any useful information, since all the differing residues are located around the active site. The close-up is too small to be useful. It would thus be more appropriate to zoom on the active site, to provide a readable picture of the substrate docked in the active site, and to provide an superposition of the active sites of the two protein highlighting differences in the proximity of the substrate.
- 3) The choice of the proteins in the alignment in figure 3B does not sound very appropriate: it would be much better to align proteins with validated LAMT activity as reference, to better highlight relevant differences with CaLAMT-like protein. Even the LAMT from *O. pumila* mentioned later in the manuscript was not used.
- 4) The molecular evolution efforts of the authors did not seem successful and, in this case, it would be better to skip. I however wonder if a proper sequence alignment has been used for the calculations, since this alignment was not provided, nor the phylogeny serving to define the background sequences (this information should be available in the sup material). The authors mention that the number of sequence available was not sufficient to provide reliable data. This is indeed possible, but I wonder if they realized that they have a huge resource to find homologs in the 1kP database. It should also be mentioned that mutations retained only in the active site would be extremely likely to occur under very strong positive selection. The selection should be relieved after the first mutation leading to activity loss if another activity did not evolve at the same time.

Concerning the two SLAS genes, their functional characterization was already reported in the literature and the only novelty is that they are issued from the WGD revealed in the first part of the manuscript. The attempt at testing the molecular evolution at work cannot be properly evaluated in the absence of the alignment and phylogeny that served as a basis for the calculations. Here too, the number and choice of the sequences used is critical, and the 1kP resource might not have been properly exploited. Note also that residues under positive or relaxed selection are interesting if spotted on the structure of the protein so as to explain the divergences in enzyme activity. After duplication, one of the genes is expected to accumulate mutations due to relaxed selection.

No information is provided concerning the two 7-DLH-like genes. The reason for their conservation is not investigated.

In conclusion very little novel, solid and convincing information is still provided by the second half of the manuscript. This information is neither displayed nor commented in an appropriate way. The authors made real efforts to improve the manuscript but these efforts were not structured and well-thought. They do not much improve the first version of the manuscript.

Reviewer #2 (Remarks to the Author):

The present revised manuscript has incorporated additional experiments and suggestions made in a previous review of the original submission.

1. A question was raised in the previous review: How did the additional genome re-sequencing improve the prospects for characterizing this putative CaLAMT? Was this because the original study did not produce a full length CaLAMT clone?.

- The authors specified that CaLAMT (caa_locus_129614_iso_1_len_1217_ver_4) was indeed full-length and that the novelty in the present report was the functional biochemical studies to show that CaLAMT possessed no LAMT activity compared with an active CrLAMT.
- The purpose of this question was to find out if additional sequencing furnished any key insights about evolution of this pathway. It seems that biochemical characterization of CaLAMT could have been carried out without the additional sequencing.
- The biochemistry produced the key take home message of this report that an inactive LAMT was the primary reason to produce secologanic acid that could then be used by strictosidinic acid synthase for assembly of strictosidinic acid precursor of camptothecin.
- The authors do point out that the present data base has a more complete set of genes that previous efforts and can be used to identify more genes in this pathway and their organization in the genome.

2. Another key issue raised in the previous review was: "Could it be that *C. acuminata* never evolved a functional LAMT?"

- While it is agreed that 'the caveats do not affect the major conclusion that CaLAMT was not functional as an authentic LAMT', suggestions that CaLAMT activity was 'functionally' lost should be removed from the text of the paper, since it is possible that authentic LAMT activity never evolved in this plant species.
- L 274-277 For example the following statement could be altered: "These results indicate that 'inactivation or functional alteration (modify)' of the LAMT orthologous (orthologues) and two highly expressed SLAS genes generated by WGD in *Cam. acuminata* may be the key reason why this medicinal plant produces different indole alkaloids by using a similar pathway with *Cat. roseus*."

3. The choice of amino acids to modify for the CrLAMT mutagenesis work.

- a. You modified 178 S-T; 261 G-C; 262 A-T; 263 G-S; 264 L-M; 266 H-F
- b. Why not 186 K-M or 186 V-I?

4. Why was a gain of function for CaLAMT not considered?

5. The reference (formerly 32) to Yang Y, et al. Bifunctional Cytochrome P450 Enzymes Involved in Camptothecin Biosynthesis. ACS Chem Biol 14, 1091-1096 (2019) has completely disappeared in the reference section of the revised manuscript. Reference 32 is now "Bathe U, Tissier A. Cytochrome P450 enzymes: A driving force of plant diterpene diversity. Phytochemistry 161, 149-162 (2019)."

Abstract:

Based on short-read sequencing data, previous genomic and transcriptomic studies revealed only a limited number of candidate genes for camptothecin biosynthesis in *Camptotheca acuminata* and the evolutionary steps leading to the camptothecin pathway remain unresolved.

Why are you using the abbreviations *Cam. acuminata* and *Cat. roseus* in the text? Simplify to *C. acuminata* and *C. roseus*.

Reviewer #3 (Remarks to the Author):

The questions have been well addressed.

REVIEWER COMMENTS

Reviewer #1 (Remarks to the Author):

NCOMMS-20-23704A

The reviewers deployed some efforts to improve the manuscript. The first part, dedicated to the *C. acuminata* genome resequencing, reads fine and seems coherent. Unfortunately the second, dedicated to the evolution of the camptothecin pathway, is still inadequate, both from a redactional and conceptual points of view.

The manuscript text is awkward and inappropriate, with meaningless sentences, ill-used English, repetitions, many mistakes that are not typos since repeated, even the corrections kindly suggested in the first round of review by Reviewer 2 have not been implemented (and there are too many to list). Gene names are not italicized, and there is a big mix-up between genes and proteins, resulting in genes with enzymatic activities. Even gene expression is mentioned as having enzymatic activity in the conclusion. When genes do not encode proteins having the same activity, they should be called homologs and not orthologs, as is the case in the manuscript. P450 families are not defined based on their activity (as mentioned in text) but on their phylogeny and based on decision of nomenclature committee (nevertheless the family labelling in sup figure 15 seems consistent and most likely appropriate). In many places the text should be shortened to remove redundant and verbose sections. What stems from previous work should be more clearly stated and not mixed up with new data provided in the manuscript to introduce ambiguity. Note also that at least some reference numbering does not seem to be correct. The authors should ask for the assistance of an experienced researcher to update their final version.

Response: We did our best to modify and rewrite the corresponding parts with likely mistakes in our revised manuscript, please see all labeled contents in the revised.

The story itself, based on the interesting statement that different pathways evolved in different species to generate the same or related compounds, is interesting, but it is not correctly exploited.

The demonstration that the CaLAMT homolog has no LAMT activity is convincing.

Though :

1) According to the information provided, all the residues differing between CrLAMT and CaLAMT-like proteins are located around the active site and all, except one, lead to the loss of LAMT activity. This is extremely unlikely to occur when a gene undergoes pseudogeneization, whereupon mutations should be dispatched on the whole sequence. So, the data provided rather hint at acquisition of new function.

Response: Not all different residues between CrLAMT and CaLAMT-like proteins are located around active sites (Supplemental Fig. 19). However, these key sites may directly affect the ability of CaLAMT-like proteins to bind loganic acid (Fig. 4A, 4B). Protein sequence alignments, phylogenetic analyses and enzyme activity experiments (Fig. 4C, 4D) suggest that the active LAMTs are split from the inactive CaLAMT when two lineages diverge. Therefore, evolutionary divergences of two lineages of LAMTs may have evolved with early speciation. It is likely that the active *LAMT* gene had become a pseudogene in *C. acuminata* or evolved into another function. However, without the ancestral state of the closely related basal group of both lineages, it is difficult to determine. Under any scenario, our results suggest that evolutionary divergence of the LAMT gene, contribute greatly to origin of camptothecin biosynthesis in *C. acuminata*.

2) In the figure 3A, showing the models of the whole CrLAMT and CaLAMT proteins does not provide any useful information, since all the differing residues are located around the active site. The close-up is too small to be useful. It would thus be more appropriate to zoom on the active site, to provide a readable picture of the substrate docked in the active site, and to provide an superposition of the active sites of the two protein highlighting differences in the proximity of the substrate.

Response: Thanks for your suggestion. We have modified them accordingly (see Fig. 4).

3) The choice of the proteins in the alignment in figure 3B does not sound very appropriate: it would be much better to align proteins with validated LAMT activity as reference, to better highlight relevant differences with CaLAMT-like protein. Even the LAMT from *O. pumila* mentioned later in the manuscript was not used.

Response: We added available *LAMT*-like genes from other species from NCBI, 1kP database and other published genomes for our final analyses (Fig. 4C).

4) The molecular evolution efforts of the authors did not seem successful and, in this case, it would be better to skip. I however wonder if a proper sequence alignment has been used for the calculations, since this alignment was not provided, nor the phylogeny serving to define the background sequences (this information should be available in the sup material). The authors mention that the number of sequence available was not sufficient to provide reliable data. This is indeed possible, but I wonder if they realized that they have a huge resource to find homologs in the 1kP database. It should also be mentioned that mutations retained only in the active site would be extremely likely to occur under very strong positive selection. The selection should be relieved after the first mutation leading to activity loss if another activity did not evolve at the same time.

Response: We added *LAMT*-like genes from other species in NCBI, 1kP database and other published genomes for sequence alignment and selection pressure analyses (Fig. 4C). Branch-site model (BSM) analyses using the *C. acuminata* *LAMT*-like gene as foreground branch showed no sites under significant positive selection, and the two-ratio branch model (BM) result showed that the strength of natural selection might have been relaxed in the foreground branches (Supplemental Fig. 20 and Supplemental Table 23-24).

Concerning the two SLAS genes, their functional characterization was already reported in the literature and the only novelty is that they are issued from the WGD revealed in the first part

of the manuscript. The attempt at testing the molecular evolution at work cannot be properly evaluated in the absence of the alignment and phylogeny that served as a basis for the calculations. Here too, the number and choice of the sequences used is critical, and the 1kP resource might not have been properly exploited. Note also that residues under positive or relaxed selection are interesting if spotted on the structure of the protein so as to explain the divergences in enzyme activity. After duplication, one of the genes is expected to accumulate mutations due to relaxed selection.

Response: We added the alignment and phylogenetic analyses (Supplemental Fig. 23-24).

No information is provided concerning the two 7-DLH-like genes. The reason for their conservation is not investigated.

Response: The two 7-DLH-like genes similarly originated from the recent WGD event specific to *C. acuminata* in the same syntenic block with the two *SLAS* genes while the two candidate 7-DLH-like genes were tandem repeat genes as for the two *SLAS* genes (Supplemental Fig. 22). We added this information in the discussion part.

In conclusion very little novel, solid and convincing information is still provided by the second half of the manuscript. This information is neither displayed nor commented in an appropriate way. The authors made real efforts to improve the manuscript but these efforts were not structured and well-thought. They do not much improve the first version of the manuscript.

Response: We provided evidence that evolutionary divergence of the *LAMT-like* gene in *C. acuminata* and positive evolution of two *SLAS* genes to converts loganic acid to secologanic acid contributed greatly to camptothecin biosynthesis in *C. acuminata*. This is a novel finding for our understanding evolutionary origin of camptothecin biosynthesis.

Reviewer #2 (Remarks to the Author):

The present revised manuscript has incorporated additional experiments and suggestions made in a previous review of the original submission.

1. A question was raised in the previous review: How did the additional genome re-sequencing improve the prospects for characterizing this putative CaLAMT? Was this because the original study did not produce a full length CaLAMT clone?.

- The authors specified that CaLAMT (caa_locus_129614_iso_1_len_1217_ver_4) was indeed full-length and that the novelty in the present report was the functional biochemical studies to show that CaLAMT possessed no LAMT activity compared with an active CrLAMT.

- The purpose of this question was to find out if additional sequencing furnished any key insights about evolution of this pathway. It seems that biochemical characterization of CaLAMT could have been carried out without the additional sequencing.

- The biochemistry produced the key take home message of this report that an inactive LAMT was the primary reason to produce secologanic acid that could then be used by strictosidinic acid synthase for assembly of strictosidinic acid precursor of camptothecin.

- The authors do point out that the present data base has a more complete set of genes than previous efforts and can be used to identify more genes in this pathway and their organization in the genome.

Response: The previous genome or transcriptome or other analyses annotated this CaLAMT gene from *C. acuminata*. But without the present high-quality genome, it is difficult to determine how many LAMT-like genes in *C. acuminata* and whether this gene has similar or changed function with other LAMT homologs. In addition, we also recover a more complete set of genes for the camptothecin pathway.

2. Another key issue raised in the previous review was: “Could it be that *C. acuminata* never evolved a functional LAMT?”

- While it is agreed that ‘the caveats do not affect the major conclusion that CaLAMT was not

functional as an authentic LAMT’, suggestions that CaLAMT activity was ‘functionally’ lost should be removed from the text of the paper, since it is possible that authentic LAMT activity never evolved in this plant species.

• L 274-277 For example the following statement could be altered: “These results indicate that ‘inactivation or functional alteration (modify)’ of the LAMT orthologous (orthologues) and two highly expressed SLAS genes generated by WGD in *Cam. acuminata* may be the key reason why this medicinal plant produces different indole alkaloids by using a similar pathway with *Cat. roseus*.”

Response: Yes, this is likely. We have made corresponding revisions. See the response to the first reviewer. The major protein sequence alignments, phylogenetic analyses and enzyme activity experiments (Fig. 4C, 4D) suggest that the active LAMTs are split from the inactive CaLAMT when two lineages diverge. Therefore, evolutionary divergences of two lineages of LAMTs may have evolved with the early speciation. It is likely that the active *LAMT* gene had become a pseudogene in *C. acuminata* or evolved into another function. However, without the ancestral state of the closely related basal group of both lineages, it is difficult to determine which is the most likely. Under any scenario, our results suggest that evolutionary divergence of the LAMT gene, contribute greatly to origin of camptothecin biosynthesis in *C. acuminata*.

3. The choice of amino acids to modify for the CrLAMT mutagenesis work.

a. You modified 178 S-T; 261 G-C; 262 A-T; 263 G-S; 264 L-M; 266 H-F

b. Why not 186 K-M or 186 V-I?

Response: The docking analyses and binding energy calculations indicate that the amino acids chosen here for modification should have seriously caused ligand binding related hydrogen bonds reduced (Fig. 4A, 4B).

4. Why was a gain of function for CaLAMT not considered?

Response: Yes. It is likely. See the above response. We revised our conclusion to ‘evolutionary divergence’.

5. The reference (formerly 32) to Yang Y, et al. Bifunctional Cytochrome P450 Enzymes Involved in Camptothecin Biosynthesis. ACS Chem Biol 14, 1091-1096 (2019) has completely disappeared in the reference section of the revised manuscript. Reference 32 is now “Bathe U, Tissier A. Cytochrome P450 enzymes: A driving force of plant diterpene diversity. Phytochemistry 161, 149-162 (2019).”

Response: We are sorry for the mistakes in reference citing. We have corrected them in the revise manuscript.

Abstract:

Based on short-read sequencing data, previous genomic and transcriptomic studies revealed only a limited

number of candidate genes for camptothecin biosynthesis in *Camptotheca acuminata* and the evolutionary steps leading to the camptothecin pathway remain unresolved.

Why are you using the abbreviations *Cam. acuminata* and *Cat. roseus* in the text? Simplify to *C. acuminata* and *C. roseus*.

Response: We have simplified all *Cam. acuminata* and *Cat. roseus* to *C. acuminata* and *C. roseus*.

Reviewer #3 (Remarks to the Author):

The questions have been well addressed.

REVIEWERS' COMMENTS

Reviewer #2 (Remarks to the Author):

The revised manuscript has been much improved and the authors have completely responded to the questions raised in the previous review of this manuscript. The documentation of the lack of LAMT activity for the *Camptotheca* LAMT-like gene and the molecular basis for this displays the major reason for the production of strictosidinic acid precursor for the subsequent assembly of camptothecin. In addition that improved genome assembly described in the present report also has improved the resource of candidate genes that will allow the complete characterization of downstream enzymes required for the assembly of this pathway.

[Editor: Reviewer #1 is unavailable. We asked Reviewer #2 to comment your responses to Reviewer #1's concerns. Reviewer #2 states in Remark to Editor section that Reviewer #1's concerns have been addressed.]